# Creation of a point-of-care therapeutics sensor using protein engineering, electrochemical sensing and electronic integration

Rong Cai [1] ✉, Chiagoziem Ngwadom [1], Ravindra Saxena[2,3], Jayashree Soman[1], Chase Bruggeman[4], David P. Hickey [4], Rafael Verduzco [3] & Caroline M. Ajo-Franklin [1,3,5] ✉

Point-of-care sensors, which are low-cost and user-friendly, play a crucial role in precision medicine by providing quick results for individuals. Here, we transform the conventional glucometer into a 4-hydroxytamoxifen therapeutic biosensor in which 4-hydroxytamoxifen modulates the electrical signal generated by glucose oxidation. To encode the 4-hydroxytamoxifen signal within glucose oxidation, we introduce the ligand-binding domain of estrogen receptor-alpha into pyrroloquinoline quinone-dependent glucose dehydrogenase by constructing and screening a comprehensive protein insertion library. In addition to obtaining 4-hydroxytamoxifen regulatable engineered proteins, these results unveil the significance of both secondary and quaternary protein structures in propagation of conformational signals. By constructing an effective bioelectrochemical interface, we detect 4-hydroxytamoxifen in human blood samples as changes in the electrical signal and use this to develop an electrochemical algorithm to decode the 4-hydroxytamoxifen signal from glucose. To meet the miniaturization and signal amplification requirements for point-of-care use, we harness power from glucose oxidation to create a self-powered sensor. We also amplify the 4-hydroxytamoxifen signal using an organic electrochemical transistor, resulting in milliampere-level signals. Our work demonstrates a broad interdisciplinary approach to create a biosensor that capitalizes on recent innovations in protein engineering, electrochemical sensing, and electrical engineering.

Since blood reflects a person's physiological profile, including nutrition[1], metabolites[2], health/disease[3,4], therapy progression[5,6], and environment[7], developing sensors to reveal biochemical information from blood is a high priority. However, there are few point-of-care (POC) devices capable of rapidly extracting information from the blood because its opacity, chemical complexity, and presence of cells create significant hurdles for sensing. Efforts to adapt the glucometer for various clinical analyses have received tremendous attention in the last decade, since an adapted glucometer might leverage existing glucometer manufacturing to develop manifold new sensors at a low

cost[8,9]. Alexandrov et al. have pioneered this direction by modulating glucose oxidation signals using engineered glucose dehydrogenase (GDH). They incorporated analyte-specific recognition proteins into GDH to detect various analytes, including rapamycin, tacrolimus, amylase, and cyclosporine A[10–12]. However, developing effective electrochemical signal transmission and further integration into self-contained devices for POC use remain outstanding challenges.

Our goal in this study is to develop a glucometer-based allosteric sensor (GBAS, Fig. 1) to electrochemically sense specific biomarkers in blood and meet the miniaturization and signal amplification requirements for POC use. Since frequent monitoring and rapid testing are critical in preventing drug resistance[13,14] or cancer recurrence[15], we chose to develop an estrogen glucometer that can respond to 4-hydroxytamoxifen (4-HT), a metabolite of the drug tamoxifen widely used in treating hormone receptor-positive breast cancer and carried in the blood[16]. In our sensor, electrical current generated from enzymatic glucose oxidation transmits presence of 4-HT via engineered allostery. After successfully modulating the glucose current, we devised an electrochemical algorithm to decode the 4-HT signal, effectively mitigating interference caused by varying glucose levels within the sample. In addition, we harnessed glucose to power the sensor, eliminating the need for bulky batteries. We further integrated this self-powered sensor with an organic electrochemical transistor (OECT) to amplify the signal with the current reaching the milliampere range, demonstrating it as an effective method to boost the change of enzymatic turnover.

## Results

### Creating GDH redox modulator with ER-LBD
As a first step in developing the GBAS, we designed a redox protein modulator that could rapidly transmit the 4-HT signal via glucose

oxidation. This was achieved by genetically fusing the ligand-binding domain (LBD) of the estrogen receptor alpha (ERα) with pyrroloquinoline quinone-dependent glucose dehydrogenase (PQQ-GDH from *Acinetobacter calcoaceticus*). We hypothesized that the 4-HT specific conformational changes in the LBD (Fig. 1B) would modulate the redox activity of GDH, leading to changes in the resulting electrical signals. In previous studies, LBD was inserted into Venus fluorescent protein[17], Cas9[18,19], aminoacyl-tRNA synthetase[20] and ferredoxin[21] to yield switches that regulate fluorescence output or cell growth. However, predicting insertion sites that do not lead to destabilization is still beyond the state-of-the-art[22,23]. In order to identify permissive sites that can introduce rapid allostery while maintaining the robustness of GDH, we created an LBD-GDH library by inserting LBD across all 454 amino acid positions of GDH using the targeted domain insertion library technique (detailed in SI Section 2.1)[24].

### Mapping the permissible and allosteric insertion sites of GDH
Deep sequencing analysis revealed that the generated library possessed a comprehensive insertion coverage (Fig. S2). We expressed this library in *E.coli* (BL21) and picked 7392 colonies, representing the entire library (>99.9% probability)[25]. We then sorted these colonies according to their ability to couple glucose oxidation to the reduction of DCPIP (2,6-dichlorophenolindophenol), an assay that is based on color change from blue to colorless[26](Figs. S3–5). In the first round, we identified 232 (~51%) insertion sites within GDH capable of catalyzing glucose oxidation in the presence of 4-HT (Fig. 2A). These active variants were further screened by comparing their rate of DCPIP reduction with 4-HT or DMSO, revealing 71 (~16%) 4-HT regulatable sites. These results show that GDH is a robust enzymatic scaffold with high domain insertion tolerance and tunability. The structure of GDH

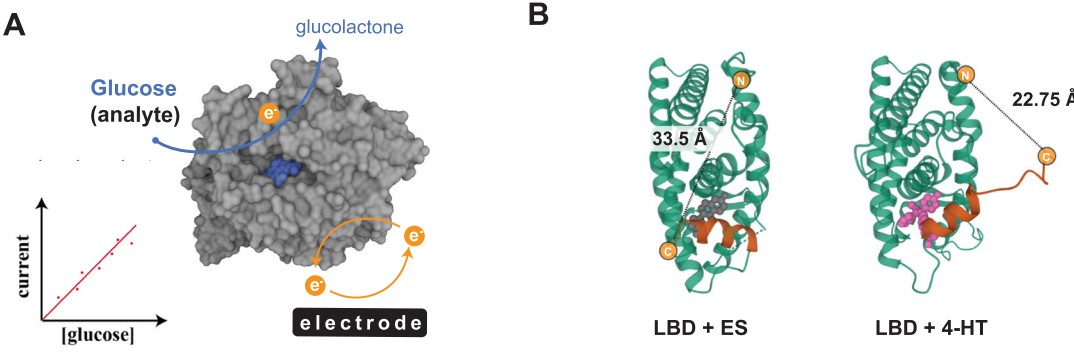

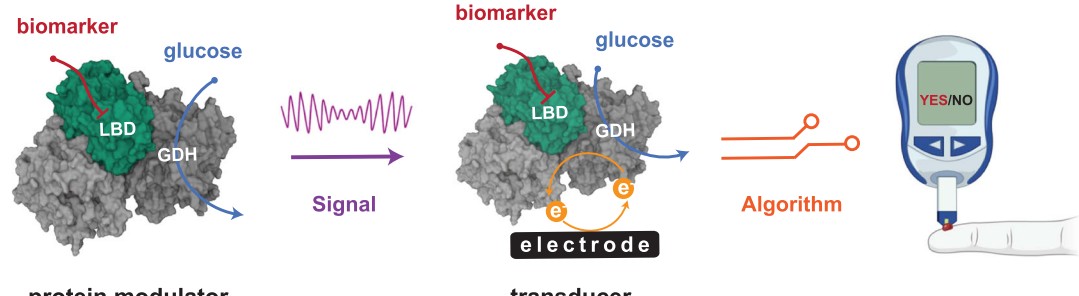

**Fig. 1 | Creation of glucometer-based allosteric sensors. A** A conventional glucometer utilizes glucose oxidase or glucose dehydrogenase (PDB: 1CRU) to catalyze glucose oxidation, generating a current proportional to the glucose concentration. **B** Crystal structure of LBD with ES or 4-HT. ER-LBD with ES (PDB: 1ERE); ER-LBD with 4-HT (PDB: 3ERT). **C** For our glucometer-based allosteric sensor, biomarker-binding signals are encoded into glucose oxidation and modulated by an allosteric GDH created by inserting a ligand binding domain (LBD) into glucose dehydrogenase (GDH). The resulting modulated signal is then transmitted and decoded by an electrochemical algorithm that generates a binary outcome. Created with BioRender.com.

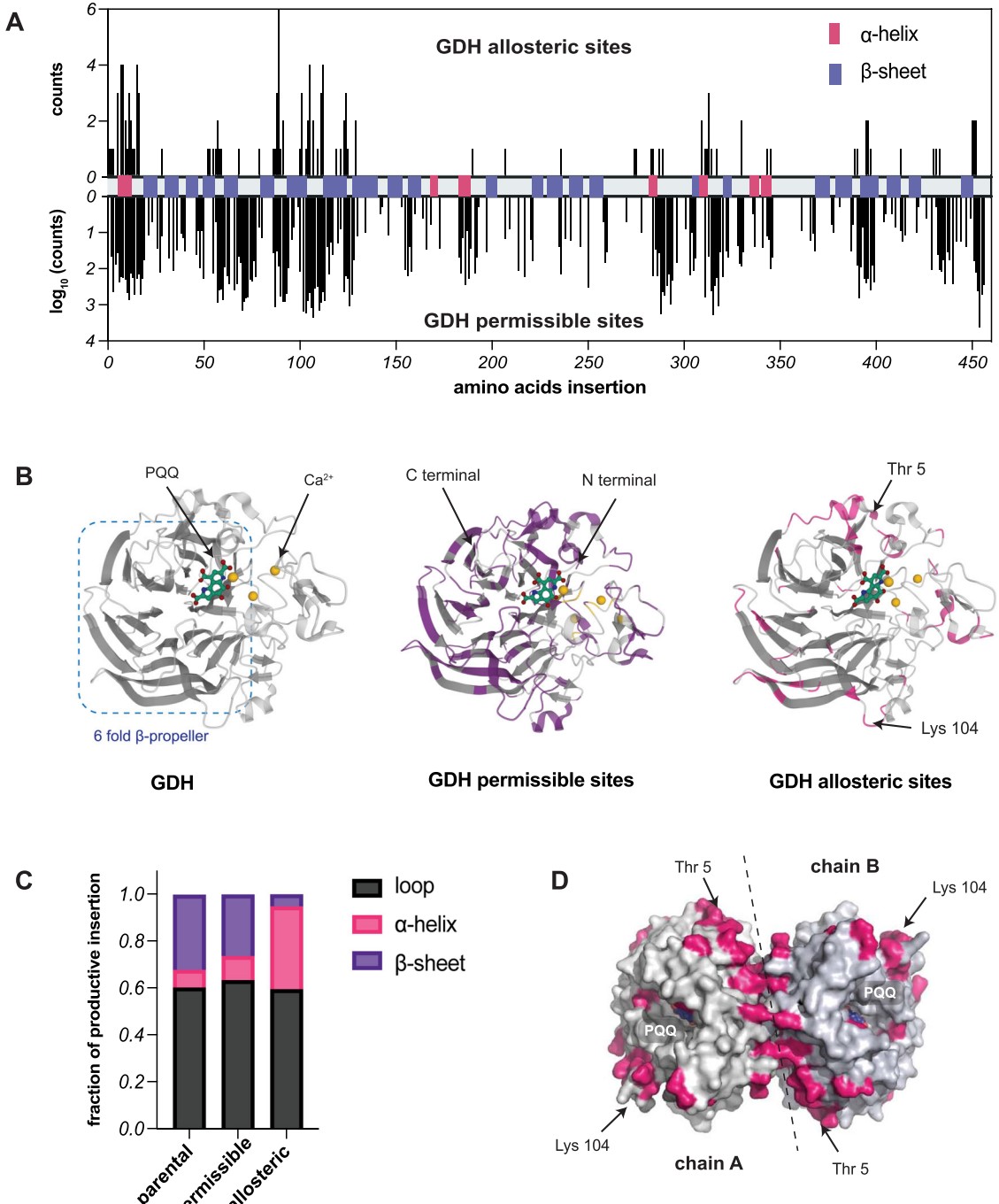

**Fig. 2 | Profiling GDH for permissible and allosteric insertion sites. A** Sequence reading counts for permissible and allosteric insertion sites were identified based on GDH activity assay. Secondary structures were highlighted along the GDH primary sequence. **B** Permissible and allosteric insertion sites mapped onto the GDH monomer structure (PDB: 1CRU). **C** The distribution of productive insertions located at loops, α-helix, and β-sheet. (**D**) Allosteric insertion sites mapped onto the GDH dimeric structure. Source data are provided as a Source Data file.

consists of a six-fold β-propeller, where a four-stranded antiparallel β-sheet is splayed outward and radially arranged around a central axis (Fig. 2B). The intrinsic stability of the β-propeller fold[27] could explain the protein's overall high tolerance to domain insertion. The circularly packed folding array is linked by loops of diverse sequences and varying lengths that carry residues essential for catalysis. Indeed, the majority of the allosteric insertion sites are situated along these flexible loops (Fig. 2C). This finding reinforces the idea that conformational flexibility is essential for signal transduction in allosteric phenomena[28]. When mapped onto the GDH homodimer structure, we

noticed that a lot of the allosteric sites were concentrated on the dimeric interface of GDH (Fig. 2D). Given the dimeric assembly is required for GDH catalysis[29], our map suggests that the local conformational change of LBD affects dimerization of GDH, consequently modulating its catalytic activity.

**Validating and improving the redox modulator GDH-5E**
Because of its outstanding allosteric response during the whole cell screening process (Fig. S6), the insertion variant GDH-5E, where LBD was inserted following residue Thr 5 of GDH, was chosen as the target

protein for further studies. After expressing and purifying the recombinant protein, we again used the DCPIP colorimetric assay to measure the catalytic activity. While wild-type GDH has no response to 4-HT, the glucose oxidation rate of the variant GDH-5E decreases by 5.5% in the presence of 1 nM 4-HT (Fig. 3A). This observation differs from the whole cell assay results, where GDH-5E displayed a more substantial change (31%) in vivo. We assume this difference is due to the transition from the *E.coli* cytosol to the buffer medium[11,30]. Since this modest effect may not be sufficient to generate a sensor, we decided to enhance the allosteric modulation on GDH-5E. Our mapping analysis revealed that flexibility is essential for signal propagation in GDH. Consequently, we posited that releasing the rigidity symmetrically around the cofactor could help the allosteric effect (detailed in SI section 2.3 and Fig S7). We further engineered GDH-5E by adding a short polypeptide linker (SGRPGSLS) after the GDH allosteric site Lys 104. Unexpectedly, the resulting variant GDH-5E⁺ oxidized glucose ~2.0 times faster than wild-type GDH (Fig. S11 and Table S2). Although DMSO showed inhibition on GDH activity (Fig. S13), the addition of 4-HT again repressed glucose oxidation, effectively attenuating the signal by 18% ($112 \pm 2$ versus $92 \pm 2$, mean ± sd, p value = 0.0005, n = 3). This allosteric inhibition happened over a wide range of 4-HT concentrations, from 338 pM to 2 µM, which is comparable to serum concentrations of tamoxifen[31] (Fig. 3B). When we evaluated the capability of GDH-5E⁺ to sense other endocrine therapeutics, we found that GDH-5E⁺ could be regulated by hexestrol, diethylstibestrol and lasofoxifene with good selectivity over 17β-estradiol, the primary female sex hormone (Fig. 3C). Thus, the allosteric variant GDH-5E⁺, designed by mapping allosteric sites on GDH, serves as an effective modulator for constructing sensors for therapeutics that exhibit rapid responses.

## Creating an amperometric sensor with GDH-5E⁺

To utilize this allosteric redox protein as part of a device, our next step was to devise a strategy to enable current flow from GDH-5E⁺ to an electrode. There are two problems associated with wiring GDH-5E⁺ on electrodes. First, the redox-active PQQ is insulated by a peptide shell, inhibiting electron transfer. Second, the protein structure and dynamics need to be preserved for signal transmission. To overcome these challenges, we immobilized GDH-5E⁺ with a redox polymer, ferrocene-modified linear poly(ethylenimine) (Fc-LPEI)[32,33]. The resulting hydrogel simultaneously provides a porous matrix to embed GDH-5E⁺ at the electrode surface and mediates electron transfer from PQQ to the electrode via the 1,1'-dimethyl ferrocene side chain (Fig. 3D). Upon addition of 10 nM 4-HT to the protein, there was a decrease in the oxidative current, confirming that immobilization preserves repression of GDH-5E⁺ by 4-HT (Fig. 3E). Having successfully interfaced GDH-5E⁺ with an electrode, we sought to test whether this system could report the 4-HT signal amperometrically, in a manner analogous to glucose sensing by glucometers. We measured electric current vs. time with a constant potential maintained at 544 mV vs. SHE (Standard Hydrogen Electrode) as mixtures of glucose with 100 µM 4-HT were injected into the stirring electrolyte solution. With successive additions of glucose, we observed steady-state current increases consistent with GDH-5E⁺ catalyzing glucose oxidation. The apo form of GDH-5E⁺ (GDH-5E⁺ containing no PQQ cofactor), in contrast, showed no response to either glucose or 4-HT. The signal from glucose/4-HT showed a lower increase over time (Fig. 3F) compared to that from glucose/17β-estradiol. The statistically significant inhibition signals appeared after ~4 min with good selectivity over 17β-estradiol (Fig. 3G). This result encouraged us to explore whether our electrochemical system could be used to test for markers in human blood. Upon addition of either 100 µM 17β-estradiol or 100 µM 4-HT to human blood samples without external glucose supplements, we observed oxidative current due to a persistent baseline concentration of glucose in the blood (average 5.6 mM in adults). Notably, blood samples with 4-HT showed a significantly lower current response (~18%) than the samples with estradiol (after background correction: $3.2 \pm 0.2$ µA vs. $3.9 \pm 0.3$ µA, p = 0.0475, Fig. 3H). Control experiments with bovine serum albumin (BSA) were also performed to account for any signal from whole blood (Fig. S16). Thus, our 4-HT GBAS is capable of rapid transmission of 4-HT signals via electrochemical glucose oxidation with remarkable selectivity even in a physiological environment.

## Developing an algorithm for binary outcome

To operate in a sample-to-result mode, the therapeutic sensor must be capable of making a Yes or No determination regarding the presence of 4-HT based on the current response. However, the current also depends on the glucose concentration in the blood sample, potentially interfering with this determination. To obtain accurate detection of 4-HT, we hypothesized that normalizing the current from GDH-5E⁺ to the current from GDH would mitigate the glucose response. We therefore analyzed the electrochemical kinetics of both GDH-5E⁺ and GDH with DMSO (blank) or 1 µM 4-HT (Fig. 4A, B). As the glucose concentration increased, the enzymatic reaction rate increased and saturated at $V_{max}$. Importantly, in the presence of 4-HT, GDH-5E⁺ exhibited reduced catalytic activity, leading to a lower $V'_{max}$, while the $V_{max}$ of GDH was unaffected. To minimize the influence of glucose on 4-HT determination, we calculated the $i_{GDH-5E+}$ to $i_{GDH}$ ratio ($i_{GDH-5E+}/i_{GDH}$) and examined it under two conditions: without 4-HT and with 1 µM 4-HT (Fig. 4C). The results revealed that 4-HT consistently diminished $i_{GDH-5E+}/i_{GDH}$ throughout the range of glucose concentrations. As glucose concentration surpasses 0.4 mM, both $i_{GDH-5E+}/i_{GDH}$ curves reach their maximum, so that changes in glucose concentration no longer have a significant effect on the ratiometric signal. Consequently, we designated glucose >0.4 mM as the intended operational range for our sensor, which should cover the vast majority of situations since the concentration of glucose in the blood normally varies between 4 and 6 mM. Within this specific range, the blank sample maintains a consistent $i_{GDH-5E+}/i_{GDH}$ ratio of $6.37 \pm 0.1$ (n = 8, Fig. 4D), while the 4-HT-containing sample remains at $4.85 \pm 0.1$ (n = 9).

Based on these findings, we sought to utilize the $i_{GDH-5E+}/i_{GDH}$ to indicate the presence of 4-HT in blind samples: a ratio near 6.37 suggests no 4-HT, while a ratio around 4.85 indicates the presence of 4-HT. To obtain both $i_{GDH-5E+}/i_{GDH}$ from a single test, we introduced an additional GDH-coated electrode into our device (Fig. 4E). After adding 200 µL of human whole blood (~5.6 mM glucose) to 2 mL of electrolytes, we determined the $i_{GDH-5E+}/i_{GDH}$. For blank blood, it was $6.1 \pm 0.5$, while for blood containing 1 µM 4-HT, it was $4.6 \pm 0.4$ (n = 3, p = 0.039, Fig. 4F). The $i_{GDH-5E+}/i_{GDH}$ observed in the blood sample was close to the ratio calculated from electrochemical kinetics data. This similarity affirms the effectiveness of our algorithm in detecting 4-HT from blind samples.

## Creating a self-powered sensor with GDH-5E⁺

To enable continuous measurements of sensing cancer therapeutics at POC locations, a prototype device must be completely portable and have a small footprint. To eliminate the need for a potentiostat or sizable batteries, we envisioned creating a self-powered sensor by incorporating GDH-5E⁺ into an enzymatic fuel cell (EFC). In a glucose/$O_2$ EFC, the oxidation of glucose delivers electrons to the anode. These electrons then travel through the external circuit to the cathode, where enzymatic reduction of dioxygen to water completes the circuit and generates power from glucose and $O_2$ (Fig. 5A). We reasoned that if 4-HT repressed glucose oxidation by GDH-5E⁺, the EFC would produce less current and less power, simultaneously acting as a biosensor and power source. Thus, we used the GDH-5E⁺/Fc-LPEI as the EFC's bioanode and paired it with a laccase (from *Trametes versicolor*) cathode that catalyzes $O_2$ reduction[34]. The open circuit potential (OCP) of our

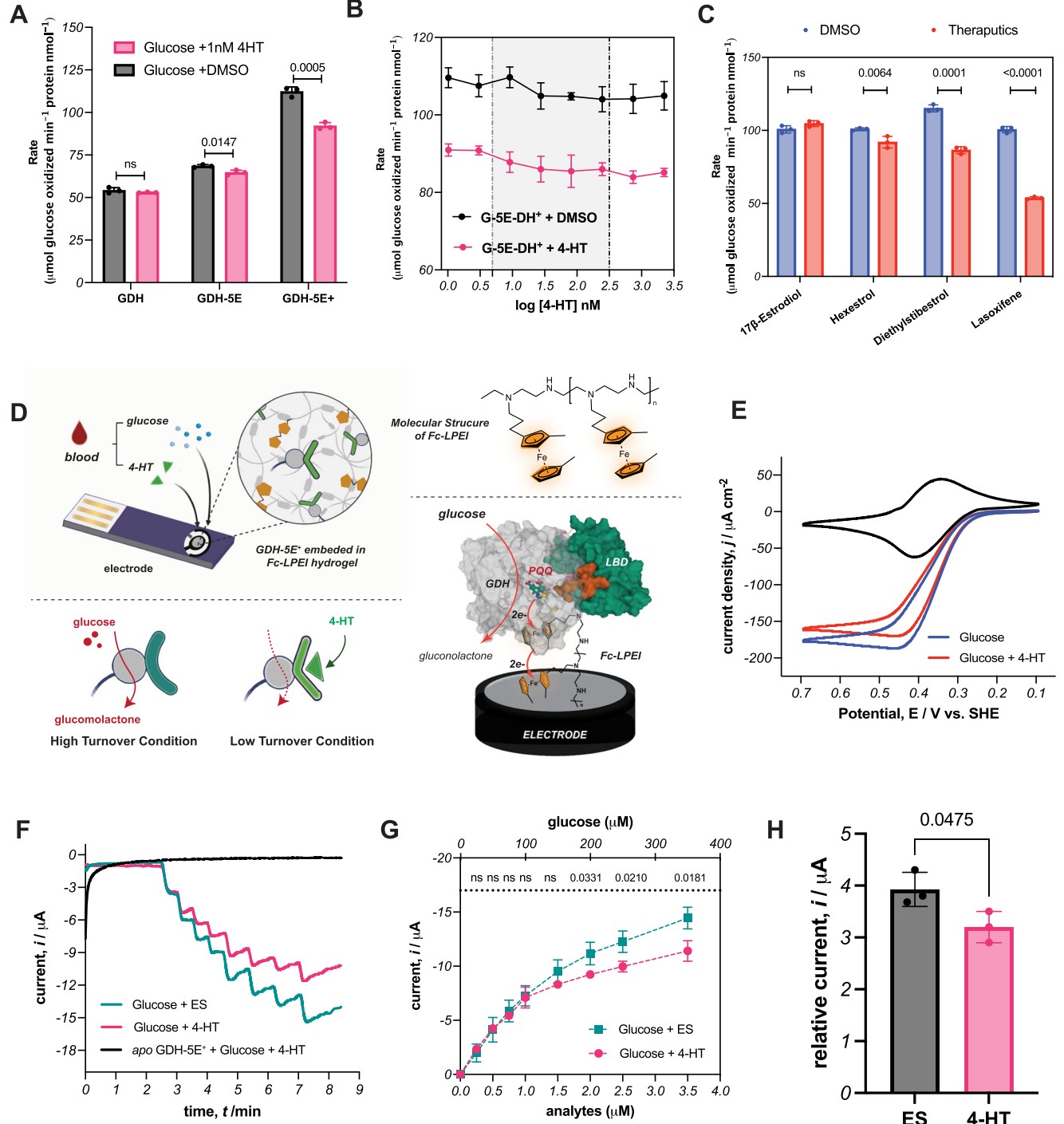

**Fig. 3 | Characterization of allosteric GDH-LBD. A** Comparing the allosteric effect of 1 nM 4-HT on GDH, GDH-5E and GDH-5E⁺ with 80 mM glucose. **B** Inhibition of 4-HT on GDH-5E⁺. The gray box highlights the clinic 4-HT concentration (5- 480 nM) in serum. **C** Allosteric effect of various endocrine therapeutics (1 nM) on GDH-5E⁺ with 80 mM glucose. **D** Scheme of wiring GDH-5E⁺ with Fc-LPEI on electrodes for 4-HT sensing. Created with BioRender.com. **E** Representative cycle voltammetry of GDH-5E⁺/Fc-LPEI with 20 mM glucose and 100 μM 4-HT. Experiments were performed at 20 mV/s with a glassy carbon electrode (diameter, 3 mm). **F** Representative amperometric i-t traces for glucose with 4-HT. A solution containing 12.5 μM 4-HT or ES and 10 mM glucose solution was added successively to 8 mL 100 mM MOPS electrolyte (pH = 7). Amperometric experiments were performed under an applied potential of 544 mV vs. SHE with AvCarb electrodes (0.25 cm²) and stirring. **G** Summary and statistical analysis of triplicated amperometric experiments in Fig. 3F. The upper X-axis represents the final concentration of glucose in the electrolytes, while the lower X-axis indicates the final concentration for either ES or 4-HT. **H** The current response to 60 μL blood with 100 μM 4-HT or ES after background correction. Values are plotted in mean ± sd with three independent experiments. Source data are provided as a Source Data file.

blood glucose/O₂ EFC is 690 ± 22 mV. When discharging this EFC, we observed lower current density when 4-HT was present in the blood sample rather than 17β-estradiol (Fig. 5B, C). The maximum current density delivered with 17β-estradiol is 179 ± 5 μA cm⁻², while it is 157 ± 6 μA cm⁻² with 4-HT. The repression by 4-HT also lowers the maximum power density: 32 ± 2 μW cm⁻² and 26 ± 1 μW cm⁻² for blood samples with 17β-estradiol and 4-HT, respectively. Therefore, incorporating the allosteric variant GDH-5E⁺ in the glucose/O₂ EFC renders

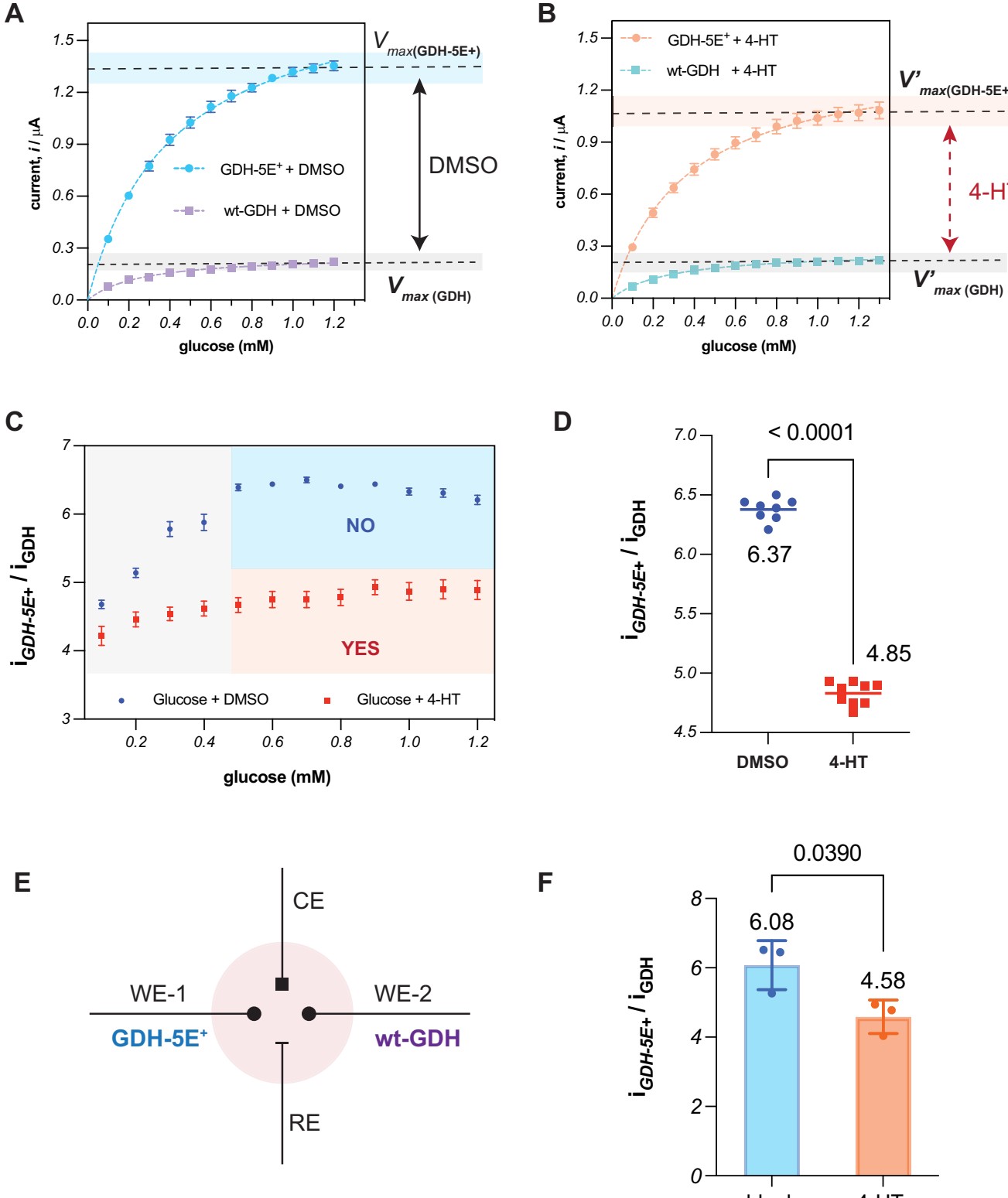

**Fig. 4 | Electrochemical algorithm allows a Yes or No determination of 4-HT.**
Comparison of the electrochemical kinetics of GDH-5E⁺ and GDH as a function of glucose concentration with DMSO (**A**) and 4-HT (**B**). Premixed 100 mM glucose with DMSO or 20 μM 4-HT added successively to 4 mL 100 mM MOPS electrolyte (pH = 7). Experiments are performed under an applied potential of 694 mV vs. SHE, with glassy carbon electrodes (diameter, 3 mm), stirring. **C** The current ratio ($i_{GDH-5E+}/i_{GDH}$) with and without 1 μM 4-HT as a function of glucose concentration. The $i_{GDH-5E+}/i_{GDH}$ values were calculated based on the data presented in Figs. 4A, B with error propagation. **D** Statistical analysis of the current ratio ($i_{GDH-5E+}/i_{GDH}$) over glucose concentrations

ranging from 0.5 to 1.2 mM. Numerical values were obtained by calculating the means from replicates ($n = 8$ for DMSO and $n = 9$ for 4-HT). **E** The equivalent electronic circuit with two working electrodes (WEs). WE-1 is coated with GDH-5E⁺, while WE-2 is coated with wild type GDH (wt-GDH). **F** The current ratio ($i_{GDH-5E+}/i_{GDH}$) for 200 μL blood (~5.6 mM glucose) and 200 μL blood with 1 μM 4-HT added to 2 mL 100 mM MOPs buffer, pH = 7. Values are plotted in mean ± sd with three independent biological experiments. Two-tailed, unpaired $t$ test for statistical analysis. Source data are provided as a Source Data file.

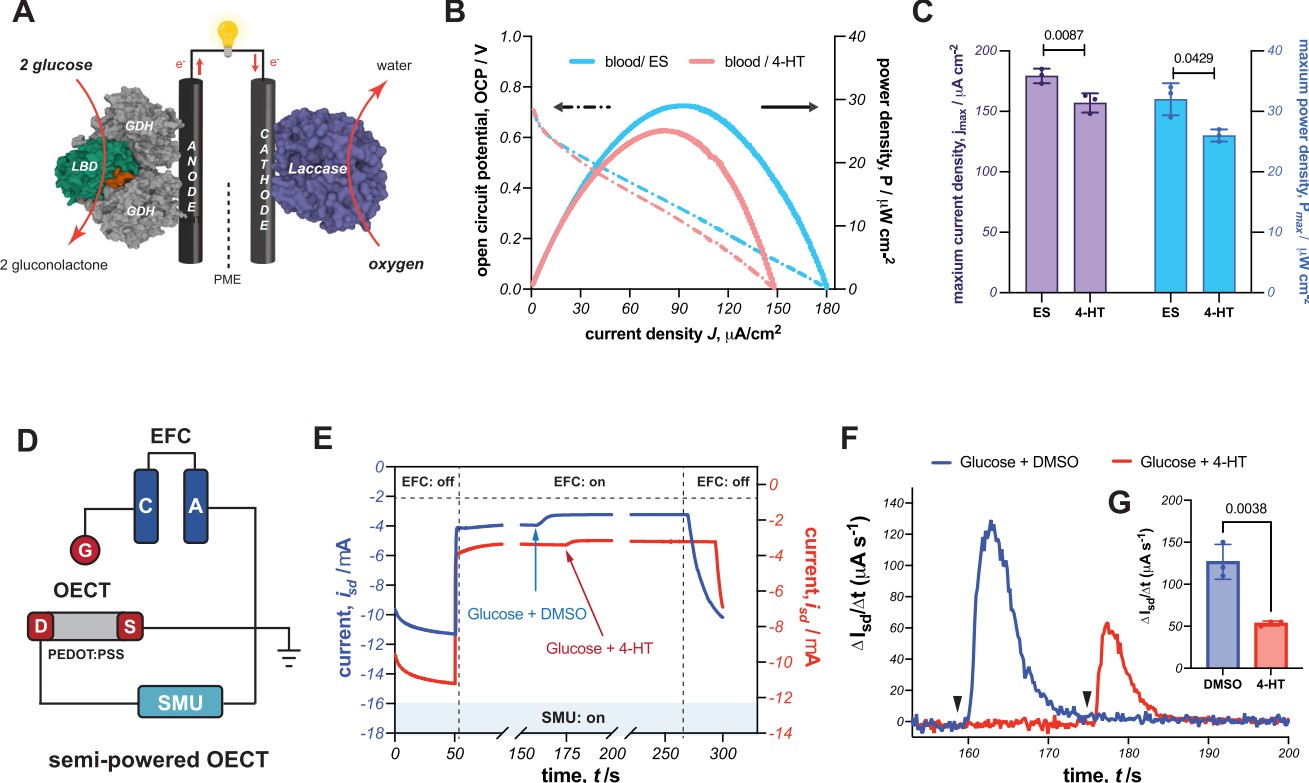

**Fig. 5 | Self-powered 4-HT sensor coupling with OECT. A** scheme of glucose/$O_2$ self-powered sensor. Created with BioRender.com. **B** Representative polarization and power curves of the EFC with 200 μL blood and 100 μM 4-HT or ES, performed linear sweep voltammetry (5 mV/s) from OCP until short circuit. **C** Summary and statistical analysis of triplicated polarization experiments. Values are plotted in mean ± sd. Two-tailed $t$ test for statistical analysis. **D** The equivalent electronic circuit for semi-powered OECT. SMU, source measure unit. **E** Representative channel current response in semi-powered OECT throughout the connection, measurement and disconnection process. 40 μL of 2 M glucose with 1 μM 4-HT or DMSO (blank) was added to the anodic chamber. **F** The representative first derivative of channel current over time for 20 mM glucose with 10 nM 4-HT. The black arrows indicate the injection time. **G** Summary and statistical analysis of triplicated OECT experiments. Values are plotted in mean ± sd. Two-tailed $t$ test for statistical analysis. Source data are provided as a Source Data file.

blood glucose as the power source as well as the signal carrier, and provides an exciting opportunity to integrate our self-powered sensor with advanced devices.

## Coupling the self-powered sensor with OECT

Low-amplitude electrochemical signals from enzymes hinder device miniaturization because electrode impedance inversely scales with size, resulting in a diminished signal-to-noise ratio. To develop a real-time, on-site biosensor for detecting 4-HT, it is crucial to amplify these weak enzymatic signals without adding complexity to the sensor circuitry. One effective solution is integrating an EFC sensor with an OECT. In this approach, the EFC sensor generates power for the gate electrode and provides a signal for the OECT to amplify. Inal et al. have demonstrated this concept with an EFC-OECT integrated glucose sensor, where the glucose concentration-dependent OCP controls the gate voltage ($V_g$) to dictate the drain current ($I_{sd}$)[35]. Thus, we reasoned that we could align an OECT in parallel with our self-powered sensor to boost the signal from 4-HT. To do so, we connected the cathode of our self-powered sensor to the gate and grounded the anode with the source of the OECT. A drain current was induced by a source meter unit (SMU) to convey and amplify the input signal from the gate, as shown in Fig. 5D and S22. The channel of this OECT was made of PEDOT: PSS that operates in depletion mode, where the application of a positive gate voltage decreases the drain current. In this configuration, the glucose EFC should provide a voltage (OCP) on the gate electrode. Indeed, when our glucose-free EFC (OCP ~ 620 mV) was connected to the OECT as described above, the drain current decreased by ~7.0 mA (Fig. 5E). As we fueled the anode with glucose, $I_{sd}$ dropped another ~0.5 mA, which is proportional to the glucose

contribution to OCP [Eq. 1].

$$\text{OCP} = K_{total} + \frac{RT}{2F} \ln \frac{C^*_{glucose}}{C^*_{gluconolactone}} \tag{1}$$

where $K_{total} = E^{\circ\prime}_{\frac{O_2}{H_2O}} + \frac{RT}{4F}\left(\frac{C^*_{O_2}}{C^*_{H_2O}}\right) - E^{\circ\prime}_{\frac{gluconolactone}{glucose}}$

However, no statistically significant difference was observed from the $I_{sd}$ generated from glucose with DMSO and glucose with 4-HT. This result aligns with our findings in the self-powered sensor, where the presence of 4-HT does not impact the OCP (Fig. S20).

Recalling that 4-HT inhibits the rate of glucose oxidation ($V_{max}$) at high glucose concentrations (Fig. 4A, B), we incorporate $V_{max}$ into Eq. 1 (see detailed mathematical derivation in SI section 6.3), which yields:

$$\text{OCP}(t) = K_{total} + \frac{RT}{2F} \ln \left( \frac{C^0_{glucose} - V_{max}t}{V_{max}t} \right) \tag{2}$$

By differentiating Eq. 2 with respect to time, we can correlate $V_{max}$ to $I_{sd}$ as:

$$\left| \frac{dI_{sd}}{dt} \right| \propto \left| \frac{d\text{OCP}}{dt} \right| \propto \left| \frac{-V_{max}}{C^0_{glucose} - V_{max}t} \right| \tag{3}$$

Based on Eq. 3, we hypothesize that the rate of decrease in drain current ($\frac{dI_{sd}}{dt}$) would be slower in the presence of 4-HT.

The first derivative of source-drain current over time is depicted in Fig. 5F, G. As we expected, 4-HT reduces the rate of change of source-drain current from $127 \pm 16\ \mu A\ s^{-1}$ to $53 \pm 2\ \mu A\ s^{-1}$ ($n = 3$;

$p = 0.0038$). Therefore, we achieved a signal modulation of 58% while generating signals in the milliampere range. More broadly, these results introduce an effective way to amplify modulator effects in glucose oxidation by integrating an OECT with a self-powered sensor.

## Discussion

One current limitation of this work is that the single EFC described here is insufficient for real-time and in-situ sensing. A single EFC cannot drive conventional transmitters, like Bluetooth (1–10 mW) and radio frequency identification (1–103 mW)[36]. However, multiple EFCs in series can provide adequate power. For example, Nishizawa et al. utilized a series of multiple fructose/$O_2$ EFC units to generate effective transdermal electroosmotic flow, facilitating glucose sampling and drug delivery with a microneedle array skin patch[37]. Alternatively, magnetic fields can be used to power devices inside the human body. Wang and Mercier made strides with magnetic human body communication (-0.4 μW), enabling efficient power-to-frequency conversion[38]. They successfully developed a self-powered, ingestible wireless biosensing capsule for real-time glucose monitoring in the porcine gastrointestinal tract. Due to the simplicity of operation, rapid signal generation, and high compatibility, we foresee our GBAS systems leveraging these technologies to evolve into skin patches or wireless biosensors in the future.

Overall, this study describes two parallel innovations that transform the conventional glucometer into a POC therapeutics sensor. First, conventional protein switch engineering depends on extensive protein structural modifications to create binary on/off signal responses. Unlike this approach, we used an engineered protein in which an analyte (4-HT) modestly modulated the rate of glucose oxidation. We developed an electrochemical algorithm to decode the 4-HT signal from the glucose signal. Second, in addition to utilizing glucose to carry the 4-HT signal, we also harnessed power from glucose oxidation to supply energy to the detection and amplification circuits. Our work showcased a self-powered POC sensor and OECT gate. The coupling between OECT and EFC sensor resulted in higher modulation, with the current reaching the milliampere range. This approach, where an OECT is coupled with a self-powered sensor, also introduces an effective methodology for reading out allosteric effects. Together, we anticipate these engineering strategies and analysis methods will stimulate a variety of GBAS capable of detecting steroid therapeutics. These sensors hold broad applications, ranging from doping controls in sports medicine to monitoring medication compliance.

## 'Methods'

The original GDH plasmid was synthesized by GenScript with codon optimization. All the derivative GDH plasmids were constructed using Q5 hot start DNA polymerase (New England Biolabs, NEB) to produce amplicons and ligate to the target sequences with Golden Gate DNA assembly. All plasmid sequences were verified with Sanger sequencing. Twist Bioscience synthesized the SPINE-generated oligo pools. Integrated DNA technologies synthesized all the primers. Single donor human whole blood was purchased from Innovative Research.

Saturated calomel electrodes (SCE), glassy carbon (GC) electrodes (diameter 3 mm), and Pt mesh (1 cm x 1 cm) were purchased from CH Instruments, Inc. AvCarb carbon paper (MGL190, untreated) was purchased from Fuel Cell Earth (USA). Ultra-high purity $O_2$ and $N_2$ were purchased from Airgas. Ethylene glycol diglycidyl ether (EGDGE) was purchased from Polysciences Inc. Poly(3,4 ethylene dioxythiophene) doped with poly(styrene sulfonate) (PEDOT:PSS, PH1000) was purchased from Heraeus Epurio Company. Water used in experiments was filtered using an Ultrapure MilliQ system. All other chemicals were purchased from Sigma Aldrich and used as received without further purification.

The utilization of human blood in this study strictly adheres to the reviewed and approved protocol (IBC-22-120) by the Institutional Biosafety Committee at the Office of Research Integrity of Rice University. Single donor blood, purchased from Innovative Research, undergoes comprehensive testing for FDA-required viral markers, yielding negative results for HBsAg, HCV, HIV-1, HIV-2, HIV-1Ag or HIV-1 NAT, ALT, West Nile Virus NAT, Zika NAT, and syphilis using FDA-approved methods. Stringent measures for personal protective equipment, including lab coats, gloves, and eye protection, were consistently implemented, and all handling occurred within a biological safety cabinet. No sharps were employed throughout the entire process. The generation of aerosols was minimal, as the experiments involved less than 5 mL of blood. Additionally, procedures such as vortexing, centrifugation, or other aerosol-producing methods were not used.

Electrochemical experiments were performed using a CH Instruments 1242 C and 1230 C potentiostat. GC electrodes were polished with a 0.05 μm alumina slurry prior to use. The OECT outputs were recorded using custom-made MATLAB software, which controls the source measure units (SMU, Keithley 2612).

## GDH-LBD library construction

We used the Saturated Programmable Insertion Engineering (SPINE) algorithm[24] to design and generate a vector library that expresses GDH with LBD inserted across the open reading frame (Fig. S1). Python-GDH plasmid (without BsaI or BsmbI recognition sites) were prepared in FASTA format, and then submitted to the custom algorithm (https://github.com/schmidt-lab/SPINE). This program generates oligo sequences, their corresponding primers for amplification, and the target backbone primers for inverse PCR (Supplementary Data 1). These oligos were microarray-synthesized and amplified as eight oligo pools. Each pool joined its backbone in parallel, then pooled together as the intermediate library. The CFU of the eight sub-libraries ranged from 1200 to 5200, corresponding to >99.99 % coverage. The Sanger sequencing confirmed that nearly 80 % of the colonies were perfect variants (Table S1). All eight libraries were pooled together at an equimolar ratio, resulting in the intermediate library with the genetic handle crossing GDH. Lastly, the LBD sequence replaced the genetic handle through BsaI-mediated Golden Gate cloning, resulting in the LBD-inserted GDH library (yielded >9200 colonies with 81.5 % perfect variants).

## Deep sequencing of GDH-LBD libraries

The plasmids from the GDH-LBD library were extracted using a Monarch Plasmid Miniprep kit. The product served as the template for specific GDH-LBD amplification with 15 cycles of PCR using the Q5 hot start polymerase. The resulting amplicon was run on 1.2% agarose gels, purified by gel extraction, and quantified by Quant-iT Picogreen. A total of 1.6 μg of DNA was prepared for deep sequencing using the Illumina MiSeq, $2 \times 150$ bp configuration by GENEWIZ. Insertion sites (Fig. S2) were identified from 140,340 raw sequencing reads. Alignments were processed using the DIP-seq pipeline (http://github.com/SavageLab/dipseq) on both forward and reverse reads.

## GDH-LBD whole cell assay

GDH-LBD library DNA was transformed into 50 μL *E. coli* BL21 chemical component cells (NEB) for protein expression. The transformed cells were diluted to various concentrations and grown on LB agar plates with 50 μg/mL kanamycin overnight at 37 °C. The following day, single colonies were hand-picked and transferred to 96 deep-well plates in 600 μL of LB medium, complementing with 10 mM $CaCl_2$, 1 μM PQQ, 50 μM IPTG and 50 μg/mL kanamycin. The cells grew at 25 °C with shaking at 60 rpm for 18 h. The final $OD_{600}$ were 0.4−0.6.

Whole-cell GDH-LBD activity was determined spectrophotometrically at room temperature by following the reduction of dichlorophenolindophenol (DCPIP) at 600 nm, using phenazine methosulfate (PMS) as a primary electron mediator (Fig. S3). A reagent

solution containing 45 mL MOPS buffer (10 mM, pH 7), 1 mL DCPIP (20 mg dissolved in 5 mL of $H_2O$ overnight), 1 mL PMS (45 mg in 5 mL of $H_2O$, freshly made and kept in the dark) and 1 mL glucose (1 M in MOPS, overnight) were prepared in each use. GDH-LBD activity was correlated to the velocity of DCPIP oxidation which was reported as the absorption decreasing at 600 nm over time. The coefficient of variance for whole cells assay was calculated as 11% (Fig. S4). Given the ideal coefficient of variance is 10%[39], our assay provided a reliable screen approach.

## GDH-LBD library screening

We first screened the GDH-LBD library for the variants capable of oxidizing glucose with 4-HT. In the colorimetric assay, the GDH-LBD active variants could turn the dark blue reagent solution yellow or colorless. We performed the screen in a Costar 96 flat transparent plate with 10 μL GDH-LBD cells mixed with a 180 μL assay reagent. Positive controls were prepared with cell expression wide-type GDH. Negative control was made with 10 μL cells harboring empty vectors to eliminate the interference of reagent decay over time (Fig. S5). By arbitrarily comparing the color decay to the control group, GDH-LBD variants showing activity within 5 min were pursued to the second screen round. All the active variants were pooled together, and their plasmids were extracted using the Monarch Plasmid Miniprep kit. The resulting plasmids were sequenced using the Illumina MiSeq, 2 × 150 bp configuration by GENEWIZ, yielding 54430 read counts. On both forward and reverse reads, alignments were processed using the DIP-seq pipeline (http://github.com/SavageLab/dipseq).

In the second round, 10 μL active GDH-LBD cells were mixed with 180 μL reagent and 10 μL DMSO or 4-HT (1 mM dissolved in DMSO). Absorption at 600 nm was recorded every 15 s for 10 min by Tecan Spark plate reader at room temperature with orbital shaking at 500 rpm. Biologically independent experiments ($n = 3$) were performed in this screening round. The data were collected and processed with two-tailed, independent $t$ tests for the $P$ value. Only those variants that showed statistically different ($P < 0.05$) activities with either 4-HT or DMSO were identified as allosteric variants. LBD-insertion sites on GDH were identified by Sanger sequencing. Their change degree is shown in Fig. S6.

## Protein purification and reconstitution

Frozen cells (-30 g) were thawed and resuspended in 150 mL buffer (50 mM HEPES, 10 mM imidazole, 300 mM NaCl, 3 mM $CaCl_2$ and 5 mM beta-mercaptoethanol, pH 7) with lysozyme and DNase, and lysed with an AVESTIN EmulsiFlex-C3 homogenizer. The lysate was clarified at 22,000 x $g$ for one hour to precipitate the cell debris. Protein is bound on a 5 mL Histrap FP Ni-NTA column (Cytiva) on an FPLC and washed with a 20% gradient of imidazole buffer (50 mM HEPES, 300 mM imidazole, 300 mM NaCl, 3 mM $CaCl_2$ and 5 mM beta-mercaptoethanol, pH 7). The protein was eluted in 40% imidazole buffer and loaded on a 15 mL HiTrap Desalting column (Cytiva) to remove imidazole. PQQ was added to the protein solution at a 2:1 molar ratio. The mixture was stirred for 30 min. Excess PQQ was removed by loading again on the desalting column. The purity of the protein was confirmed by SDS−PAGE (Fig. S8), and the concentration was measured with Bradford assay. The reconstitution of PQQ was verified using UV-Vis spectroscopy (Figs. S9 and 10).

## GDH-LBD protein activity

A reagent solution containing 47 mL MOPS buffer (10 mM, pH 7), 1 mL DCPIP (20 mg dissolved in 5 mL of $H_2O$), 1 mL PMS (45 mg in 5 mL of $H_2O$, freshly prepared and kept in the dark) were prepared for each use. The assay was performed on a flat transparent 96 plate in triplicates. Since GDH has a high turnover rate, each test requires only 5−10 ng of protein. The experiments were prepared by mixing 10 μL protein with a 180 μl reagent solution. Reactions were initialized by

adding 10 μL glucose and recorded every 15 s for 10 min by Tecan Spark plate reader with orbital shaking at 500 rpm. (The coefficient of variance is 7.6 %)

## Fc-LPEI synthesis

Fc−LPEI was synthesized as previously reported[32,33]. High molecular weight polyethylenimine (LPEI, 0.100 g, 2.33 mmol) was dissolved in a mixture of acetonitrile (7 mL) and methanol (3 mL) at 80 °C. A solution of 3-bromopropyl-dimethylferrocene (0.137 g, 0.45 mmol) in acetonitrile (2 mL) was added to the stirring LPEI solution at 80 °C, and the mixture was stirred at 90−100 °C for 24 h. The reaction mixture was cooled to room temperature and the solvent was removed under reduced pressure. Excess ferrocene was removed by soaking the resulting polymer in diethyl ether (10 mL) for 1 h at room temperature. The diethyl ether was decanted, and excess solvent was removed under reduced pressure. The final product was a brown malleable solid (0.169 g, 71% yield)[1].H-NMR in $CD_3OD$: δ (ppm) 1.67 (2H, C-CH₂-C), 1.94 (6H, Fc(-CH₃)), 2.30 (2H, Fc-CH₂-C), 2.50 - 3.10 (16H, N-CH₂-C, polymer backbone), 3.87 (7H, Fc-H).

## Protein/Fc−LPEI hydrogel film preparation

Nonwet-proofed AvCarb carbon paper electrodes were cut into strips (3 cm × 0.5 cm). Except for the designed electrode surface area, the rest of the strip was dipped into melting paraffin wax to seal the conductive surface. Before electrochemical testing, the exposed end was coated with protein/Fc-LPEI hydrogel for analysis, while the waxed end was used as an electrochemical connection point.

Six μL GDH-5E⁺ (16 mg/mL) or BSA (20 mg/mL) was added to 14 μl of Fc−LPEI solution (12 mg/mL in $H_2O$). The mix was vortexed before adding EGDGE (0.75 μL, 4.4% by volume in $H_2O$). The resulting mixture (-20 μL) was then drop-coated onto a glassy carbon electrode (diameter, 3 mm) in 3 μL aliquots or AvCarb carbon electrode (0.5 cm × 0.5 cm) in 10 μL aliquots. The resulting electrodes were cross-linked at 4 °C for 6 h.

## Cyclic voltammetry (CV)

CVs were performed with three electrodes: glassy carbon or AvCarb coated with protein/Fc−LPEI as the working electrode, a SCE as the reference, and a platinum mesh counter electrode. Experiments were performed at room temperature in 5 mL MOPS buffer (100 mM, pH 7.0). All the protein hydrogel-coated electrodes were electrochemically conditioned with 10 cycles of scanning at 100 mV/s. The electrochemical activities of GDH-5E⁺ were measured by CV scan at 20 mV/s in a stationary solution for three cycles. The third cycle was used for analysis and reporting. Reported potentials were referenced to the standard hydrogen electrode (SHE) by adding 244 mV to the measured values.

## Amperometric i−t curves

A fixed potential of 300 mV vs. SCE was first applied to working electrodes for 100 s. This step could effectively decrease the double−layer charging current in the following tests. Electrodes were soaked in a stirred solution of 100 mM MOPS, pH 7. Glucose was injected into the electrolytes after the current stabilized (usually 180 s).

## Laccase cathode preparation

Laccase from *Trametes versicolor* (≥0.5 μ/mg) was purchased from Sigma−Aldrich. This enzyme deposition solution was prepared by suspending laccase (30 mg) in 75 μl of 200 mM citrate−phosphate buffer (pH 4.5), followed by the addition of 7.5 mg of anthracene−modified multi-walled carbon nanotube (An-MWCNTs)[40]. The mixture was subjected to successive vortex/sonication steps until the ink was homogenous. TBAB-Nafion (25 μL)[34] was added, and a few more vortex/sonication steps were undertaken to promote thorough mixing. 30 μL ink was spread onto the AvCarb electrodes (1.25 cm ×

0.8 cm) and dried at 4 °C overnight. The performance of the laccase cathode was characterized with CV and shown in Fig. S17.

## Construction of the glucose/O₂ EFC

EFCs were set up in a custom-made electrochemical cell, where a Nafion® 212 proton exchange membrane (PEM) was used to compartmentalize the anodic and cathodic chambers (Fig. S18). In the anodic chamber, we use GDH-5E⁺/Fc–LPEI coated AvCarb (0.5 cm × 0.5 cm) as the anode, with 4 mL 100 mM MOPS as electrolytes. On the cathodic sides, we used laccase/An-MWCNTs/TBAB-Nafion coated AvCarb electrode (1.25 cm × 0.8 cm). This cathodic chamber contained 4 mL 200 mM citrate–phosphate buffer, pH 4.5, having O₂ bubbled.

## Fabricating OECT device

The OECT devices were fabricated using a low-cost methodology. Glass slides were first cleaned with soap and DI water, then washed in isopropanol and acetone before blow-drying with clean, dry air. Later, thermal evaporation deposited Au (65 nm) on the cleaned glass slides, which were cut into glass strips (1.2 cm × 2.5 cm). Using a new single–edge razor blade, we cut a line along the glass strip, creating a channel with a length of 50 μm. By masking with Kapton tape, we defined a 1 mm width for the channel. Additional areas where Au was deposited were left uncovered to provide electrical contact for alligator clips. The slides were further subject to UV-Ozone treatment for 10 min to ensure effective adhesion to the OECT solution. An Ag/AgCl wire was prepared as the gate electrode. The channel of OECT was made of poly (3,4 ethylene dioxythiophene) doped with poly(styrene sulfonate) (PEDOT:PSS). An OECT solution was prepared by ultrasonicating PEDOT:PSS with ethylene glycol (5 % v/v) and (3-glycidyloxypropyl) trimethoxysilane (GOPS, 1% w/w). 4-dodecylbenzenesulfonic acid (DBSA, 0.1% v/v) was added to ensure uniform mixing. This OECT solution was spin-coated to the prepared channel at 1000 rpm for 45 s and accelerated at 200 rpm/s. The resulting devices were annealed at 1400 °C for 30 min. The transfer characteristic of the OECT was shown in Fig. S21.

## Coupling self-powered sensor with OECT

We connected the SMU to power the source-drain channel, as shown in Fig. S22. The OECT was held at a constant $V_{sd}$ of − 600 mV while monitoring the $I_{sd}$. We let the current stabilize for 50 s before connecting the cathode to the gate of OECT. After $I_{sd}$ reached a steady state, 40 μL of 2 M glucose comprising 1% (v/v) 4-HT (10 μM, in DMSO) was added to the anodic chamber. The current was constantly monitored, allowing sufficient time to stable before disconnecting from EFC.

## Statistics and reproducibility

The dimensions of the library screen were predetermined in advance, as outlined in SI section 2.2.1. For all other sample sizes, a minimum of three independent biological samples or triplicate measurements across independent experiments were utilized. Statistical significance was assessed through unpaired two-tailed $t$ tests. Randomization and blinding procedures were employed to mitigate bias during experimentation. All collected data were included in the analyses without exclusion.

## Reporting summary

Further information on research design is available in the Nature Portfolio Reporting Summary linked to this article.

## Data availability

All data supporting the findings of this study are available within the article and its supplementary files. Any additional requests for information can be directed to, and will be fulfilled by, the corresponding authors. Source data are provided with this paper.

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

## Acknowledgements

Cancer Prevention and Research Institute of Texas, grant # RR190063 (to C.M.A.-F., supporting R.C., C.N. and J.S.). NSF National Research Traineeship in Bioelectronics, grant # 1828869 (supporting C.N.). Research was sponsored by the Army Research Office and was accomplished under Grant Number W911NF-22-1-0239 (to C.M.A.-F. and R.V., supporting R.S.). This material is based upon work supported by the National Science Foundation under Grant No. EFRI – 2223678 (to C.M.A.-F. and R.V.). We thank Prof. Joff Silberg and Dr. Ian Campbell for helpful conversations and the kind gift of the ER-LBD plasmid.

## Author contributions

R.C., C.N. and C.M.A.-F. have designed the project. R.C and C.N. constructed and screened the LBD-inserted GDH library. R.C. and J.S. worked on protein structure analysis, simulation, and characterization. C. B. and D.P.H. synthesized the Fc-LPEI polymer. R.C., R.S. and R.V. designed the EFC-OECT coupling, conducted the experiment and derived the equations.

## Competing interests

R.C., C.N. and C.M.A.-F. have submitted an invention disclosure, entitled "A soluble PQQ-GDH based domain insertion permissibility map." R.S., C.B., J.S., D.P.H, and R.V. declare no competing interests.

## Additional information

[1]Department of Biosciences, Rice University, Houston, TX, USA. [2]Applied Physics Graduate Program, Smalley-Curl Institute, Rice University, Houston, TX, USA. [3]Department of Chemical and Biomolecular Engineering, Rice University, Houston, TX, USA. [4]Department of Chemical Engineering and Materials Science, Michigan State University, East Lansing, MI, USA. [5]Department of Bioengineering, Rice University, Houston, TX, USA. ✉e-mail: rc82@rice.edu; cajo-franklin@rice.edu

