## [Peer Review File · Nature Communications]

REVIEWER COMMENTS

Reviewer #1 (Remarks to the Author):

On the premise of protein engineering, the authors of the present manuscript wish to create a self-powered and real-time sensor for therapeutics. However, since the present sensor is a POCT rather than a wearable, it cannot achieve "real-time" sensing because there must be a "delay" between the blood sampling time and the test time. In addition, I did not observe any data about therapeutics application. Seriously, I don't think the sensor can measure the 4-HT level in a blind blood sample since there is no reference signal (same glucose level but no 4-HT) from the same blood sample. How can we tell whether the signal acquired from the blind sample is diminished (the presence of 4-HT) or unchanged (no 4-HT) if there is no reference signal? In my opinion, this work is an early investigation of the idea, and it falls short of what authors said it achieved. Thus, the manuscript cannot be accepted for possible Nature Comm. Publication.

Reviewer #2 (Remarks to the Author):

This is a very nice study demonstrating in very elegant way how to modulate GDH activity via allosteric binding of 4HT to the enzyme via a ligand binding domain that was selected from a library and further improved its modulation range via a rational-designed protein engineering. The new recombinant enzyme was now wired to an Enzymatic Fuel Cell, powered by glucose and generate current which is modulated by 4HT.

The paper is well written and worth publication provided that the authors will clarify or modify the text with respect to the following comments:

1. It is not clear to me how natural variations of glucose levels in the blood will affect the current modulation by 4HT. I was not convinced that the different response rate of the sensor to glucose compared with 4HT will allow to separate the signals across the natural glucose and 4HT levels in the blood. It is expected that an experiment will be conducted to demonstrate the claim. Otherwise, the sensor maybe useless.
2. The authors describe a self-powered sensor but it is not clear to me if the power generated and the signal modulation is enough to transmit the information by RF from an implant to an external device. This is critical since otherwise the self-power property is not that useful if the information is wired transdermally. External devices maybe powered by batteries.

Reviewer #3 (Remarks to the Author):

Cai and colleagues present a tour de force in interdisciplinary biosensor engineering that covers the full range from the design, screening and testing of a new sensor protein to its integration into a quite complex setup of an enzyme fuel cell-powered organic electrochemical transistor (OECT). It is very rare to find this level of biochemical and device design integrated in a single study and I commend the authors for the breadth of their work. Having said that, abstract, introduction and discussion of their manuscript contain unnecessary exaggerations and could do a better job of putting this work into context. I'll dive into that in more detail below. This should not distract from the fact that the work itself is very interesting.

Overview

Technically, the study falls into three parts:

(1) Protein sensor design; Cai et al. insert the estrogen receptor α (ER α) into the structure of PQQ-dependent glucose dehydrogenase (PQQ-GDH). ER α binds the study's main target analyte, 4-hydroxytamoxifen (4-HT), leading to a conformational change which is meant to then modify the catalytic activity of GDH. ER α insertion has previously been shown to modify other enzyme activities (e.g. Cas9). PQQ-GDH is used in commercial glucose sensors and the K. Alexandrov lab has pioneered domain insertion into PQQ-GDH as a way to convert regular glucometers into biosensors for several other targets. However, the ER α - PQQ-GDH combination is novel and the structural mechanism differs from more complex "Alexandrov designs". Moreover, Cai et al. employ a sophisticated and only recently published method that screens through a protein library of all possible insertion points to extract the one with the best sensor response. This alone is impressive work although there are some technical questions which I will raise in my detailed comments below. For example, the authors improve the sensor through an additional seemingly arbitrary protein modification the rationale of which is not explained.

(2) Integration of ER α -PQQ-GDH into a regular amperometric (glucometer) setup. This follows standard practice but the choice of immobilization and electrochemical coupling is not trivial.

(3) Integration of the amperometric sensor with a glucose-powered fuel cell as an electrical input and a OECT for amplification of the output signal; There is a recent precedent for glucose fuel cell / OECT coupling (their reference 34). Nevertheless, it is a non-standard setup and an achievement to get it working. I do have doubts that the whole fuel-cell / OECT setup really improves the analyte signal though. The authors make it difficult to compare the results between the different device integration steps and a proper performance characterization (with replicates and error bars) of the final device is lacking. There is actually a supplemental figure S29 possibly suggesting that the final device cannot at all distinguish 4-HT from DMSO but, as this figure is nowhere referenced in the main manuscript, I don't really know what to make of it. More about these issues in my detailed comments below.

Manuscript shortcomings

There is an, in my view, unprofessional (and unnecessary) tendency to oversell their work. The abstract is technically uninformative and reads like a mediocre press release. Terms like “revolutionary” and “innovative” should be avoided. Worse is the author’s idea to liken their sensor with the 5G network (“modelled after the 5G network”, “we envisioned a new strategy that mimics radio access networks”, and Fig 1). 5G is a network communication standard, dealing with multi-frequency multiplexing, radio cells, error correction etc., none of which find any equivalence in this project. What the authors actually refer to is the simple modulation of a carrier wave by another signal – which is plain 103-year-old radio technology. Furthermore, their claim implies that the hybrid ligand recognition -> enzyme activity readout architecture is without precedent. That is of course not the case and the authors themselves quote several studies following the same basic scheme (and many more could be referenced). Abstract, Introduction and Discussion of the paper need a thorough cleanup from any such distracting claims and Fig. 1A should be simply removed.

Along the same lines, the authors fail to discuss any shortcomings of their design and device. In fact, I see no discussion in the conventional sense. Their current device setup is still very far from being applicable “in blood” (compare Fig. S23) and although the enzyme fuel cell can drive the OECT, readout is realized with a Keithley measurement unit which, along with the laptop, doesn’t quite look as if it would be powered by glucose just yet. That’s totally what I would expect at this early stage of development but then the authors should moderate their language. Instead of suggesting they already have a self-powered device operating in blood, they should discuss how their current prototype would enable such a future development. Moreover, the authors, several times, refer to their “glucometer-based allosteric sensor platform”. Glucometers run on batteries so what’s the advantage of the enzyme fuel cell and OECT if that’s the readout unit anyway? The authors make little attempt to place their work into the context of other studies (not a single literature reference in the whole discussion section) even though I am sure there would be a couple of interesting lessons to discuss.

Unrelated but also concerning the whole manuscript, instead of reporting mean \pm standard error of the mean, the authors’ error bars in plots should report mean \pm standard deviation. It is not clear to me what kind of error measure is used in the text but also there I would argue it should always be standard deviation rather than SEM, unless the authors clearly discuss that sensor results can only be interpreted after averaging over multiple measurements.

Many supplemental figures, text sections and tables are nowhere referenced in the main manuscript (e.g. Fig. S5, S7, S9, ...) and are left to the reader to be “discovered” without context. The same is true for subfigures 1B, 2A and 3E in the main manuscript.

Detailed / Technical comments and questions in order of appearance

* [L65] No references are given for the ER α ligand-binding domain (LBD), in particular regarding the conformational changes that are critical for sensor function.

* [L67] PQQ-GDH-based sensors from the Alexandrov lab are referenced but only α -Amylase and Cyclosporine A are mentioned as previous targets. The same lab also published Calcium and Rapamycin / Tacrolimus sensors based on the same architecture. This needs to be mentioned. Moreover, the authors claim that previous sensors were slow with a “~30 min” response time [L70]. However, looking at Guo / Alexandrov et al. 2019 (JACS), detection times were around 3 to 12 min with signals often clearly visible before that. The authors should therefore either mediate their claim or provide enzymatic turnover numbers to which their own enzyme variants can be compared.

* [L77] The description of their domain insertion library construction and screening leaves out some important details: It needs to be clarified that the library was screened in *E. coli*. Moreover, current language implies they used cell sorting (a.k.a. FACS) [L79] which they did not. They performed plate-based assays on a very large number of individual colonies (BTW, picked by hand or automated?).

* [L83] The reference to Fig. 1A likely refers to Fig. 2A (of which there is no mentioning)

* [L100] In their initial screen, the authors identify an insertion variant exhibiting a ~30% change in the rate of glucose oxidation after addition of 4-HT (Fig. S6). After large-scale purification, they then characterize this same variant again but find it has only a 5.5% decrease in glucose oxidation rate (“modestly regulated”). This discrepancy needs to be explained. In addition to the % change of rate observed in the initial screen (Fig. S6), the authors should report the absolute rates in an additional supplemental figure or, better, table. Interestingly, in their Fig. S11 / Table S2, V_{max} for this variant (GDH-5E) is reported to be 60% above the wild type GDH. I would ask the authors to repeat this kinetic analysis in the presence of the 4-HT “inhibitor” (e.g., at the same constant concentration of 1 nM used in Fig 3A) to get a better idea about how the insertion affects enzyme activity with and without target. Some discussion on how insertions can increase enzyme activity would be good.

* [L106] Based on the suddenly poor performance of their original insertion candidate, the authors quite magically decide to improve catalytic efficiency and target response of this variant by inserting a short peptide sequence elsewhere in the protein to, in fact, dramatic effect. This variant GDH-5E+ has a 2-fold increased catalytic rate which decreases by 35% in presence of 4-HT. However, it is completely unclear how and why the authors arrived at this modification. Is there pre-existing literature on improving PQQ

GDH activity by insertions? Is this based on data from their own initial screen (see my request for these data above)? What's the rational and assumed mechanism behind this design choice? The authors mention a GDH allosteric site but do not provide any reference for it.

* Fig 3A and Fig 3C (also 3H) are somewhat misleading in that their Y-axis is starting at ~40 rather than 0 and the Y-axis scales are different even though the same type of data is reported. This exaggerates relative differences between target and negative control.

* Fig 3C provides "primary sensor response", that is, GDH activity after addition of different analytes. However, there is a large variation in the negative control (DMSO) signal which suggests that there is a much larger background variance in enzyme activity than reported by the individual error bars. Again, the actual reduction in GDH activity (~20%?) appears inconsistent with the ~40% change in specific rate reported in Fig 3A. Moreover, the absolute activity with 4-HT is around 65 U/mmol in Fig. 3A but increases to 80 to 85 U/mmol in Fig 3C. Compared to the overall range of negative control values (~95 – 115?), the response to 4-HT may not look that large any longer. However, this assumes that negative control reactions are independent from target reactions. Are they? The authors should clarify where this variance comes from and how it affects their conclusions.

* [L122] The actual chemical structure of the conducting hydrogel / immobilization scheme is difficult to make out in Fig 3D. Please modify it or include a sub-figure with chemical structures showing how the proteins are chemically attached (or embedded).

* [L138] Is Estradiol used as a negative control? Is it a good negative control? Why should we not expect any conformational change from estradiol binding (as the data suggests)? Related:

[Fig 3F] I am missing the amperometric behavior with only glucose. These data should already be available as a similar experiment is shown in Fig 3G.

* [L160] Is the difference between negative control and target signal of the self-powered sensor still significant when the error is expressed as standard deviation instead of SEM?

* [L160/Fig. 4B] The authors should provide a response versus analyte concentration "calibration curve" for the fuel cell setup just as they have given it for the amperometric setup (Fig. 3G). Without such a plot, it is difficult to impossible to compare how the enzyme fuel cell architecture affects the final sensor response.

* [L184/Fig. 4E] Again, we need a [analyte] versus sensor response calibration curve (in replicate with error bars / individual points) in order to truly evaluate the performance of the enzyme fuel cell / OECT device; preferably both in the ideal buffer and in blood. The current Fig. 4E only shows the result of a single experiment, at a single concentration, which is anecdotal evidence at best.

* [L186] The authors are not the first to integrate enzyme fuel cell and OECT (see ref. 34). The claim of “new methodology” should be either explained in more detail or dropped.

* [L190/Discussion] What’s really missing is a fair comparison of amperometric, fuel-cell-powered, and fuel-cell+OECT results in the style of the Fig. S20 b box plot (BTW, not referenced anywhere and sub-plot b has no description in its legend).

* [L312] “wide-type” -> wild type

* [L325] The sentence is incomplete (grammar-wise) and incomprehensible overall.

In summary, the most serious issues relate to the presentation of the manuscript. There are some important technical questions and the merit of the final device architecture is not clearly shown. In my view, these need to be addressed before a final decision on publication can be made. However, the study stands out in covering an unusually wide and multi-disciplinary scope of engineering and in combining excellent protein design work with sophisticated device development.

Raik Gruenberg

REVIEWER COMMENTS

Reviewer #1 (Remarks to the Author):

On the premise of protein engineering, the authors of the present manuscript wish to create a self-powered and real-time sensor for therapeutics. However, since the present sensor is a POCT rather than a wearable, it cannot achieve "real-time" sensing because there must be a "delay" between the blood sampling time and the test time.

We agree with the Reviewer's comments that our therapeutic sensor prototype aligns more with a Point-of-Care Testing (POCT) device rather than a wearable sensor. The confusion arose from our initial understanding of 'real-time,' which we interpreted as implying no signal delay during testing, such as incubation time. We had not realized that 'real-time' was being associated with 'wearable'. To avoid further misunderstanding, we have removed the term 'real-time' from both the title and the text.

In addition, I did not observe any data about therapeutics application. Seriously, I don't think the sensor can measure the 4-HT level in a blind blood sample since there is no reference signal (same glucose level but no 4-HT) from the same blood sample. How can we tell whether the signal acquired from the blind sample is diminished (the presence of 4-HT) or unchanged (no 4-HT) if there is no reference signal?

We greatly appreciate the Reviewer's valuable feedback here, which was echoed by Reviewers 2 and 3. To address this concern, we have developed an algorithm specifically designed to provide a 'yes' or 'no' response regarding the presence of 4-HT, independent of glucose concentration. Detailed information about this algorithm can be found in the revised manuscript in the section titled "Developing an Algorithm for Binary Outcome" and a new Figure 4. Briefly, we introduced an additional working electrode coated with GDH and recorded the currents generated by the GDH-5E+ (4-HT regulatable GDH) and the wild-type GDH. Because 4-HT only reduces the maximum current of GDH-5E+, the current ratio $i_{\text{GDH-5E+}}/i_{\text{GDH}}$ in blank and 4-HT-containing samples becomes notably distinct. Therefore, we utilized the current ratio $i_{\text{GDH-5E+}}/i_{\text{GDH}}$ to determine the presence of 4-HT in blind samples.

In my opinion, this work is an early investigation of the idea, and it falls short of what authors said it achieved. Thus, the manuscript cannot be accepted for possible Nature Comm. Publication.

Reviewer #2 (Remarks to the Author):

This is a very nice study demonstrating in very elegant way how to modulate GDH activity via allosteric binding of 4HT to the enzyme via a ligand binding domain that was selected from a library and further improved its modulation range via a rational-designed protein engineering. The new recombinant enzyme was now wired to an Enzymatic Fuel Cell, powered by glucose and generate current which is modulated by 4HT. The paper is well written and worth publication provided that the authors will clarify or modify the text with respect to the following comments:

We thank the reviewer for their strong endorsement of our work and pointing out areas where the manuscript can be clearer.

1. It is not clear to me how natural variations of glucose levels in the blood will affect the current modulation by 4HT. I was not convinced that the different response rate of the sensor to glucose compared with 4HT will allow to separate the signals across the natural glucose and 4HT levels in the blood. It is expected that an experiment will be conducted to demonstrate the claim. Otherwise, the sensor maybe useless.

We thank the reviewer for their insightful comments, which have significantly improved the quality of the manuscript. Based on these comments (which were echoed by Reviewers 1 and 3), we have added a section titled "Developing an algorithm for binary outcome" and Figure 4.

To investigate the potential interference of glucose among various samples, we have now introduced an additional working electrode coated with GDH. Electrochemical kinetic experiments were then carried out using GDH-5E⁺ as well as the wild type GDH electrodes with both blank and 4-HT-containing solutions over a wide range of glucose (Fig. 4A and B). We also normalized the current of GDH-5E⁺ to GDH to mitigate the effect from glucose (Figure 4C). Under low glucose concentrations (0 - 0.4 mM), it is evident that the current ratio $i_{\text{GDH-5E}^+}/i_{\text{GDH}}$ is primarily determined by glucose concentration. As pointed out by the Reviewers, distinguishing the 4-HT signal from glucose in this range presents a challenge for blind samples. However, when the glucose concentration exceeds 0.4 mM, the current ratio $i_{\text{GDH-5E}^+}/i_{\text{GDH}}$ reaches maximum values, rendering changes in glucose concentration ineffective, and 4-HT becomes the predominant factor influencing the current. As a result, we have designated ≥ 0.4 mM glucose concentration as the intended operational range for the sensor. This conveniently aligns with our experimental setup where 200 μL blood (~ 5.6 mM glucose) is added to 2 mL electrolytes. Most importantly, the enzymatic reactions in these experiments exhibit saturation behavior (Fig 4F). Reporting the reaction rate within the substrate saturation range represents an effective method for filtering out the carrier signal in this type of sensor.

2. The authors describe a self-powered sensor but it is not clear to me if the power generated and the signal modulation is enough to transmit the information by RF from an implant to an external device. This is critical since otherwise the self-power property is not that useful if the information is wired trans-dermally. External devices maybe powered by batteries.

We apologize for the misleading claims about the ‘self-powering’ and ‘real-time’ properties of our sensor. We have corrected it in the revised manuscript. In our study, the power generated is only enough to drive the fuel cell type POC sensor and the gate electrode in the OECT prototype sensor. Our prototype design is insufficient to power a wearable sensor in situ.

However, our LBD-GDH holds the potential to achieve this goal with additional electronic circuit design or integration in series. For example, Joseph Wang and Patrick P. Mercier's group have reported a self-powered ingestible wireless biosensing system in a porcine model for real-time in situ glucose monitoring in the gastrointestinal tract. (Nature Communications | (2022) 13:7405 1.) Their capsule incorporates a glucose biofuel cell to sense and power an energy-efficient magnetic human body communication (mHBC). Additionally, Matsuhiko Nishizawa's group has employed a series-connected array of fructose/O₂ fuel cells to generate sufficient transdermal electroosmotic flow through a microneedle array patch (Nature Communications | (2021) 12:658). Given that our LBD-GDH shows comparable activity to their GOx or fructose dehydrogenase, substituting their protein in these systems to convert glucose sensors into 4-HT sensors with concurrent power supply will likely be a fairly trivial undertaking. Overall, our study encoded the conformational signal into glucose oxidation via molecular engineering and electrochemically decoded the conformational signal. We foresee this GDH-based engineering being useful in glucose sensor research and industrial applications. Again, we sincerely appreciate your question and include this discussion in the paper's discussion section.

Reviewer #3 (Remarks to the Author):

Cai and colleagues present a tour de force in interdisciplinary biosensor engineering that covers the full range from the design, screening and testing of a new sensor protein to its integration into a quite complex setup of an enzyme fuel cell-powered organic electrochemical transistor (OECT). It is very rare to find this level of biochemical and device design integrated in a single study and I commend the authors for the breadth of their work. Having said that, abstract, introduction and discussion of their manuscript contain unnecessary exaggerations and could do a better job of putting this work into context. I'll dive into that in more detail below. This should not distract from the fact that the work itself is very interesting.

We thank the reviewer's strong endorsement of our work and for pointing out areas in which the manuscript can be more accurate and clearer. We greatly appreciate the Reviewer's time and effort, as their comments have led to strengthening of this manuscript.

Overview

Technically, the study falls into three parts:

(1) Protein sensor design; Cai et al. insert the estrogen receptor α (ER α) into the structure of PQQ-dependent glucose dehydrogenase (PQQ-GDH). ER α binds the study's main target analyte, 4-hydroxytamoxifen (4-HT), leading to a conformational change which is meant to then modify the catalytic activity of GDH. ER α insertion has previously been shown to modify other enzyme activities (e.g. Cas9). PQQ-GDH is used in commercial glucose sensors and the K. Alexandrov lab has pioneered domain insertion into PQQ-GDH as a way to convert regular glucometers into biosensors for several other targets. However, the ER α - PQQ-GDH combination is novel, and the structural mechanism differs from more complex "Alexandrov designs". Moreover, Cai et al. employ a sophisticated and only recently published method that screens through a protein library of all possible insertion points to extract the one with the best sensor response. This alone is impressive work although there are some technical questions which I will raise in my detailed comments below. For example, the authors improve the sensor through an additional seemingly arbitrary protein modification the rationale of which is not explained.

Thank you for your insightful comments. We have revised the manuscript to include our rationale in introducing the linker to create the GDH-5E+ variant in SI section 2.3. We also summarize this rationale in response to detailed 'point o' (see below).

(2) Integration of ER α -PQQ-GDH into a regular amperometric (glucometer) setup. This follows standard practice but the choice of immobilization and electrochemical coupling is not trivial.

We appreciate the reviewer's strong endorsement of our work.

(3) Integration of the amperometric sensor with a glucose-powered fuel cell as an electrical input and a OECT for amplification of the output signal; There is a recent precedent for glucose fuel cell / OECT coupling (their reference 34). Nevertheless, it is a non-standard setup and an achievement to get it working. I do have doubts that the whole fuel-cell / OECT setup really improves the analyte signal though. The authors make it difficult to compare the results between the different device integration steps and a proper performance characterization (with replicates and error bars) of the final device is lacking. There is actually a supplemental figure S29 possibly suggesting that the final device cannot at all distinguish 4-HT from DMSO but, as this figure is nowhere referenced in the main manuscript, I don't really know what to make of it. More about these issues in my detailed comments below.

We thank the Reviewer for their constructive comments. Because the EFC-coupled OECT reads out changes in potential while the glucometer measurement detects changes in current, it is difficult to directly compare their device performance. Therefore, we have removed our statement concerning the signal-to-noise amplification in the amperometry sensor and the power in the EFC. Likewise, we have removed the un-referenced supplemental figure S29.

Manuscript shortcomings

*a) There is an, in my view, unprofessional (and unnecessary) tendency to oversell their work. The abstract is technically uninformative and reads like a mediocre press release. Terms like "revolutionary" and "innovative" should be avoided.

Thank you for your constructive comments. We have modified our abstract to remove non-technical descriptors of the work.

*b) Worse is the author's idea to liken their sensor with the 5G network ("modelled after the 5G network", "we envisioned a new strategy that mimics radio access networks", and Fig 1). 5G is a network communication standard, dealing with multi-frequency multiplexing, radio cells, error correction etc., none of which find any equivalence in this project. What the authors actually refer to is the simple modulation of a carrier wave by another signal – which is plain 103-year-old radio technology. Furthermore, their claim implies that the hybrid ligand recognition -> enzyme activity readout architecture is without precedent. That is of course not the case and the authors themselves quote several studies following the same basic scheme (and many more could be referenced). Abstract, Introduction and Discussion of the paper need a thorough cleanup from any such distracting claims and Fig. 1A should be simply removed.

We appreciate this feedback and have removed references to the 5G network and corresponding figure panels.

*c) Along the same lines, the authors fail to discuss any shortcomings of their design and device. In fact, I see no discussion in the conventional sense. Their current device setup is still very far from being applicable "in blood" (compare Fig. S23) and although the enzyme fuel cell can drive the OECT, readout is realized with a Keithley measurement unit which, along with the laptop, doesn't quite look as if it would be powered by glucose just yet. That's totally what I would expect at this early stage of development but then the authors should moderate their language. Instead of suggesting they already have a self-powered device operating in blood, they should discuss how their current prototype would enable such a future development.

*d) Moreover, the authors, several times, refer to their "glucometer-based allosteric sensor platform". Glucometers run on batteries so what's the advantage of the enzyme fuel cell and OECT if that's the readout unit anyway? The authors make little attempt to place their work into the context of other studies (not a single literature reference in the whole discussion section) even though I am sure there would be a couple of interesting lessons to discuss.

This manuscript was transferred from sister journal and, therefore was not formatted with a Discussion per Nature Communication guidelines. We have corrected this in the revised manuscript. The new discussion section includes references to work on a self-powered ingestible wireless biosensing system for real-time glucose monitoring in the gastrointestinal tract (Nature Communications | 2022; 13:7405) and a microneedle array patch self-powered by a series of fructose/O₂ fuel cells (Nature Communications | 2021; 12:658).

Additionally, we apologize for the non-precise use of the words "self-power" and "in blood" in the title, abstract and manuscripts. We have revised the text to remove these words. We appreciate your corrections.

*e) Unrelated but also concerning the whole manuscript, instead of reporting mean +- standard error of the mean, the authors' error bars in plots should report mean +- standard deviation. It is not clear to me what kind of error measure is used in the text but also there I would argue it should always be standard deviation rather than SEM, unless the authors clearly discuss that sensor results can only be interpreted after averaging over multiple measurements.

In the original manuscript, we described our data using the mean ± standard deviation (SD). We used the standard error of the mean (SEM) for error bars in our graphs, in accordance with recommendations from Nature Methods (Krzywinski, M., Altman, N. Error bars. *Nat Methods* 10, 921–922 (2013)). Recognizing that most readers will expect error bars to represent SD, we have replotted all graphs in manuscript to display error bars representing the standard deviation.

*f) Many supplemental figures, text sections and tables are nowhere referenced in the main manuscript (e.g. Fig. S5, S7, S9, ...) and are left to the reader to be "discovered" without context. The same is true for subfigures 1B, 2A and 3E in the main manuscript.

We appreciate your suggestion, and we have now referenced all the figures and tables in the updated manuscript appropriately

Detailed / Technical comments and questions in order of appearance

*g) [L65] No references are given for the ER α ligand-binding domain (LBD), in particular regarding the conformational changes that are critical for sensor function.

Thank you for your advice. We added a new Figure 1B to illustrate the conformational change of ER α ligand-binding domain (LBD).

*h) [L67] PQQ-GDH-based sensors from the Alexandrov lab are referenced but only α -Amylase and Cyclosporine A are mentioned as previous targets. The same lab also published Calcium and Rapamycin / Tacrolimus sensors based on the same architecture. This needs to be mentioned. Moreover, the authors claim that previous sensors were slow with a “~30 min” response time [L70]. However, looking at Guo / Alexandrov et al. 2019 (JACS), detection times were around 3 to 12 min with signals often clearly visible before that. The authors should therefore either mediate their claim or provide enzymatic turnover numbers to which their own enzyme variants can be compared.

We agree that this additional work from the Alexandrov group should be referenced and have included this new reference. The Reviewer is correct that Alexandrov and co-workers have built tripartite sensors, with GDH and calmodulin as constant parts, to detect analytes ranging from rapamycin, tacrolimus, amylase, and cyclosporine with detection times around of these multiple sensors range between 3 to 12 minutes. However, these assays were carried out after incubating and reconstructing their protein complex at 25 °C for 30 minutes without glucose. This information is in the aforementioned paper (Guo / Alexandrov et al. 2019 (JACS)) in the Supplementary Information and figure legend. Glucose was then subsequently added after this incubation period to measure activity. In contrast, our approach is more direct; we (1) circumvent the use of calmodulin and (2) eliminate the 30 min required for re-folding in advance of sensing.

*i) [L77] The description of their domain insertion library construction and screening leaves out some important details: It needs to be clarified that the library was screened in *E. coli*. Moreover, current language implies they used cell sorting (a.k.a. FACS) [L79] which they did not. They performed plate-based assays on a very large number of individual colonies (BTW, picked by hand or automated?).

We appreciate the Reviewer pointing out these areas that lacked clarity. In the “Mapping the permissible and allosteric insertion sites of GDH” section, we specify that the library was screened in *E. coli* (BL21). The manuscript states that we used a colorimetric assay instead of FACS (L81). In the Methods section, we provided detailed information about the manual selection of colonies for the GDH-LBD whole-cell assay.

*j) [L83] The reference to Fig. 1A likely refers to Fig. 2A (of which there is no mentioning)

Thank you for pointing this out. We have corrected it.

*k) [L100] In their initial screen, the authors identify an insertion variant exhibiting a ~30% change in the rate of glucose oxidation after addition of 4-HT (Fig. S6). After large-scale purification, they then characterize this same variant again but find it has only a 5.5% decrease in glucose oxidation rate (“modestly regulated”). This discrepancy needs to be explained.

The Reviewer makes a very perceptive point. We believe this discrepancy arises from the difference in environmental conditions, specifically the shift from the *E. coli* cytosol to the assay or electrolyte medium. Although often described as an aqueous medium, the cytosol significantly differs from the diluted aqueous buffer used for in vitro assay; the cytosol has a much higher concentration of ions, metabolites, and proteins. (PLoS Computational Biology 7, e1002066 (2011)). Additionally, the crowded environment likely fosters numerous interactions between the recombinant protein GDH-5E and native components of *E. coli*. This shift in environmental conditions could also affect the diffusion and availability of 4-HT, causing discrepancies in allosteric effects. In fact, similar discrepancies have been observed in previous GDH work (Journal of the American Chemical Society 138, 10108-10111 (2016)) as well as split protein biosensor

assays (Drug Discovery Today 21, 415-429 (2016). We have added this discussion and reference in “Validating and improving the redox modulator GDH-5E section”.

*l) In addition to the % change of rate observed in the initial screen (Fig. S6), the authors should report the absolute rates in an additional supplemental figure or, better, table. Interestingly, in their Fig. S11 / Table S2, V_{max} for this variant (GDH-5E) is reported to be 60% above the wild type GDH. I would ask the authors to repeat this kinetic analysis in the presence of the 4-HT “inhibitor” (e.g., at the same constant concentration of 1 nM used in Fig 3A) to get a better idea about how the insertion affects enzyme activity with and without target. Some discussion on how insertions can increase enzyme activity would be good.

In the initial whole cell screen, colonies were manually picked and grown for 12 to 16 hours. We did not normalize the OD_{600} or record the OD_{600} in this process, so to avoid day-to-day differences, we do not report absolute rates.

The Reviewer is asking how insertion affects enzyme activity with and without target. This question does not have a straightforward answer because of the inhibitory effects on GDH by DMSO. In the absence of target (4-HT), we conducted the MMK assay to assess V_{max} and K_M (Figure S11 and Table S2) using a DMSO-free buffer. To probe GDH activity in the presence of 4-HT, we have to dissolve 4-HT in DMSO. Control experiments show that DMSO exerts an inhibitory effect on GDH, as evidenced in the graph on the right (Figure S13, see right). Due to the inhibition effect of DMSO, the specific activity of GDH in MMK experiments (Figure S11 and Table S2) cannot be directly compared to the GDH apparent activity in Figures 3A, B, and C.

*m) [L106] Based on the suddenly poor performance of their original insertion candidate, the authors quite magically decide to improve catalytic efficiency and target response of this variant by inserting a short peptide sequence elsewhere in the protein to, in fact, dramatic effect. This variant GDH-5E+ has a 2-fold increased catalytic rate which decreases by 35% in presence of 4-HT. However, it is completely unclear how and why the authors arrived at this modification.

We appreciate the Reviewer pointing out the opportunity to explain our reasoning behind the short peptide insertion. We hypothesized that the allosteric regulation in the GDH-5E variant was due to lack of flexibility in the protein, preventing a larger conformational change between the 4-HT bound and unbound form. Thus, we sought to enhance allosteric sensitivity by increasing flexibility opposite the site of the LBD insertion (see Figure S7 below in response to point o) via this short linker. While this change yielded the larger allosteric modulation, the observed increase in catalytic rate was unexpected. We are currently investigating the origin of the increase in catalytic rate. Detailed discussions on this unexpected phenomenon are provided in response to point o).

n) Is there pre-existing literature on improving PQQ-GDH activity by insertions?

To the best of our knowledge, there is no literature about how insertion improves PQQ-GDH activity.

*o) Is this based on data from their own initial screen (see my request for these data above)? What's the rational and assumed mechanism behind this design choice? The authors mention a GDH allosteric site but do not provide any reference for it.

Figure S7. The rationale for designing GDH-5E+. (a) Hypothetical allosteric propagation pathway in PQQ-GDH. (b-d) Hypothetical split structures of GDH-5E at 104, 89 or 207.

Our screen unveiled specific locations for LBD insertion within GDH that enabled the resulting protein to be regulated by 4-HT. Since these locations were located within flexible regions in GDH (loops), these data suggested that conformational flexibility was important to allow allosteric changes in structure upon 4-HT binding (refer to Figure 2C). Building upon this discovery, our objective was to amplify the allosteric effect by augmenting flexibility by incorporating a flexible linker at previously identified allosteric positions identified through screening. We assumed the existence of an allosteric propagation pathway that traverses via Thr 5 and PQQ. We extended this potential allosteric structural rearrangement and incorporated residues 89, 207, and 104 into it (see Figure S7a). In selecting the most suitable site, we hypothesized that two criteria would facilitate the propagation of conformational signals within the protein: (1) the even spacing of components and (2) the presence of most allosteric sites at the splitting interface. We depicted the

theoretically completely split configurations of GDH-5E at positions 104, 89, or 207 (refer to Figure S7 b-d). As these structures illustrate, splitting at positions 5 and 89 does not yield evenly distributed components. In contrast, compared to site 204, breaking at site 104 exposes more allosteric sites, particularly on the back of the N-terminus. Consequently, we opted to insert the flexible linker at Lys104. We have provided a detailed description of this rational design process in the supplementary information as Section 2.3 titled “The rationale for designing GFH-5E+” and Figure S7. However, we want to clarify that we did not conduct any experiments or comparisons to validate the mechanism motivating this design. Therefore, we are uncomfortable discussing it extensively in the manuscript.

*p) Fig 3A and Fig 3C (also 3H) are somewhat misleading in that their Y-axis is starting at ~40 rather than 0 and the Y-axis scales are different even though the same type of data is reported. This exaggerates relative differences between target and negative control.

We followed the Reviewer's suggestion and re-plotted most data to enhance clarity. However, we retained some data on the zoomed scale to ensure readers could observe the standard deviations.

*q) Fig 3C provides “primary sensor response”, that is, GDH activity after the addition of different analytes. However, there is a large variation in the negative control (DMSO) signal which suggests that there is a much larger background variance in enzyme activity than reported by the individual error bars. Again, the actual reduction in GDH activity (~20%?) appears inconsistent with the ~40% change in the specific rate reported in Fig 3A. Moreover, the absolute activity with 4-HT is around 65 U/mmol in Fig. 3A but increases to 80 to 85 U/mmol in Fig 3C. Compared to the overall range of negative control values (~95 – 115?), the response to 4-HT may not look that large any longer. However, this assumes that negative control reactions are independent from target reactions. Are they? The authors should clarify where this variance comes from and how it affects their conclusions.

The Reviewer makes a keen observation here. We acknowledge that there can be significant variation between tests on different days due to the necessity of using an extremely low protein concentration (10 uL of a 1 ng/uL protein dilution derived from a 20 ug/uL stock). Such low concentrations are needed to stay within the linear detection range and calculate the initial rate. To mitigate potential errors stemming from dilution or pipetting, we measure the response from one endocrine therapeutic and one DMSO procedure simultaneously using the same diluted protein solution. In this way, while there may be variations in absolute activity across different tests, the impact of endocrine therapeutics remains evident.

The reviewer highlights an inconsistency between Figure 3C and Figure 3A. Upon reviewing the totality of our data, we see that a 20% reduction in catalytic rate was most representative of our results. We have corrected Figure 3A to reflect these observations. Thank you for bringing this to our attention.

*r) [L122] The actual chemical structure of the conducting hydrogel/immobilization scheme is difficult to make out in Fig 3D. Please modify it or include a sub-figure with chemical structures showing how the proteins are chemically attached (or embedded).

This is a good point. We have included the molecular structure of Fc-LPEI structure in Figure 3D and clarified that GDH-5E+ is embedded in the Fc-LPEI hydrogel in the manuscript.

*s) [L138] Is Estradiol used as a negative control? Is it a good negative control? Why should we not expect any conformational change from estradiol binding (as the data suggests)? Related:

Yes, estradiol is used as a negative control. This is because there is no atomically resolved structure for ER-LBD in the absence of a ligand. There are atomic resolution structures of ER-LBD with estradiol and 4-HT bound, showing a large conformational change between these structures (Fig .1B). Consequently, we designed our approach around the anticipated conformational change in an LBD-GDH fusion between ES and 4-HT binding. Thus, we carried out our tests using ES as a negative control. In addition, previous

research where ER-LBD was inserted into ferredoxin has also demonstrated no significant difference in activity between DMSO and ES (Nature Chemical Biology 15, 189-195 (2019), Figure 3D).

*t) [Fig 3F] I am missing the amperometric behavior with only glucose. These data should already be available as a similar experiment is shown in Fig 3G.

We appreciate the Reviewer's desire to see the glucose-only data alongside the other controls. This data is provided here. However, we have chosen not to depict this data in Figure 3G because we cannot overlay the data from the different experiments effectively. (This is partially because injection of additional glucose is added not according to a fixed time interval but when the current stabilizes, yielding varying injection intervals). In the revised manuscript, we have included multiple representative graphs illustrating the raw data (Figure 3F, 5B, 5E and 5F). Additionally, we provide data averaged from triplicate experiments and use statistical methods to report the statistical significance of these data (Figure 3G, 5C and 5G).

*u) [L160] Is the difference between negative control and target signal of the self-powered sensor still significant when the error is expressed as standard deviation instead of SEM?

Yes, it is. We have revised the manuscript and added the statistical assessment of differences in the control and target groups as Figure 5C.

*v) [L160/Fig. 4B] The authors should provide a response versus analyte concentration "calibration curve" for the fuel cell setup just as they have given it for the amperometric setup (Fig. 3G). Without such a plot, it is difficult to impossible to compare how the enzyme fuel cell architecture affects the final sensor response.

The Reviewer is requesting additional calibration curves to quantitatively compare the EFC-OECT device to the amperometric device. This is challenging to do because of the different readouts of the devices. More critically, our purpose of demonstrating the self-powered sensor is to illustrate that our bifunctional GDH-5E+ enzyme retains the capabilities of both LBD and GDH within the EFC configuration. Consequently, LBD-GDH has the potential to replace GDH in various applications.

*w) [184/Fig. 4E] Again, we need a [analyte] versus sensor response calibration curve (in replicate with error bars / individual points) in order to truly evaluate the performance of the enzyme fuel cell / OECT device; preferably both in the ideal buffer and in blood. The current Fig. 4E only shows the result of a single experiment, at a single concentration, which is anecdotal evidence at best.

As indicated in our response to point v, we do not seek to compare these different devices. As requested, we have added the statistical summary as Figure 5G.

*x) [L186] The authors are not the first to integrate enzyme fuel cell and OECT (see ref. 34). The claim of "new methodology" should be either explained in more detail or dropped.

The Reviewer raises an important question, and we have revised the manuscript to more clearly describe our new scientific contribution here. We present a first quantitative model that describes how changes in GDH activity in an EFC-OECT integrated device yield changes in the difference in source-drain current (dI_{sd}/dt). This model is derived from combining Michaelis-Menten enzyme kinetics, electrochemical descriptions of a fuel cell, and existing models for source-drain currents in OECTs. This model is described in a new section 6.3 in the SI, entitled "Mathematical model of self-powered sensor coupled to OECT."

Comparison of this work to prior work (including ref 34) shows that our detection and reporting method represents a novel approach.

*y) [L190/Discussion] What's really missing is a fair comparison of amperometric, fuel-cell-powered, and fuel-cell+OECD results in the style of the Fig. S20 b box plot (BTW, not referenced anywhere and sub-plot b has no description in its legend).

We understand the Reviewer's point here. However, our intention is not to claim signal-to-noise (S/N) amplification in moving from the EFC to EFC-OECD configuration. Rather, we seek to demonstrate the reporting of 4-HT under EFC-OECD configurations and believe Fig 5F and G show this.

*z) [L312] "wide-type" -> wild type

We appreciate the correction.

* aa)[L325] The sentence is incomplete (grammar-wise) and incomprehensible overall.

We appreciate the Reviewer pointing out where the manuscript is unclear. We have revised this sentence accordingly.

In summary, the most serious issues relate to the presentation of the manuscript. There are some important technical questions and the merit of the final device architecture is not clearly shown. In my view, these need to be addressed before a final decision on publication can be made. However, the study stands out in covering an unusually wide and multi-disciplinary scope of engineering and in combining excellent protein design work with sophisticated device development.

Raik Gruenberg

Dear Dr. Gruenberg,

We appreciate your kind words, and constructive feedback, and are deeply grateful for the time and effort you have dedicated to reviewing our paper. You have raised many salient points, and we hope that we have addressed them in this revised version. We hope you understand the challenges we faced in addressing certain aspects of protein engineering, particularly in reconciling the differences in relative protein activity between the whole cell assay and the protein assay (point **k**) and discussing the mechanisms behind changes in allosteric sensitivity (points **m**, **o**). We put forth significant experimental effort to answer these questions before re-submitting our manuscript, but there are places where our current knowledge remains limited. Given the circumstances, our approach is to follow your sage advice: acknowledge these limitations honestly and maintain a softer tone. Thank you again.

REVIEWERS' COMMENTS

Reviewer #1 (Remarks to the Author):

Acceptance as it is.

Reviewer #3 (Remarks to the Author):

The authors have thoroughly revised their manuscript and, where necessary, have provided new experimental data to address some of the issues raised. I believe the result is ready for publication and will find an interested readership.

I have only three suggestions for *very* minor improvements:

(1) 2nd paragraph of Introduction (Main): "To encode the signal from 4-HT in the blood, we followed the approach of Alexandrov..."

I would argue the Alexandrov work is now already perfectly referenced in the first paragraph and Cai et al.'s strategy of inserting a hormone receptor domain into PQQ-GOx is sufficiently novel and distinct to not constitute the same approach. Perhaps the authors could instead briefly mention (again) the protein engineering work / building blocks as this is IMO an important innovative aspect.

(2) page 4 and several places and figures later mention "the apo form of GDH-5E+" but I couldn't find any explanation what is missing between apo and holo enzyme. I am pretty sure, it is the PQQ co-factor we are talking about here but it would be good to spell it out at the first mentioning.

(3) Legends for Fig 3 and 4 could use a little bit more detail in order to be self-standing. In Fig 3G, the glucose concentration axis running parallel to analyte concentration is probably the final concentration due to repeated addition but that's not entirely clear from "Summary and statistical analysis".

Fig. 4C shows glucose crossed out in one section of the graph the meaning of which is not explained (without any glucose addition? Then "4-HT only" or "4HT w/o Glc" would, in my view, be clearer).

I congratulate the authors to this very nice work.

REVIEWERS' COMMENTS

Reviewer #1 (Remarks to the Author):

Acceptance as it is.

We thank the reviewer for the strong endorsement of our work.

Reviewer #3 (Remarks to the Author):

The authors have thoroughly revised their manuscript and, where necessary, have provided new experimental data to address some of the issues raised. I believe the result is ready for publication and will find an interested readership.

I have only three suggestions for *very* minor improvements:

We thank the reviewer's strong endorsement of our work and highlight the aspects that could further improve our manuscript.

(1) 2nd paragraph of Introduction (Main): "To encode the signal from 4-HT in the blood, we followed the approach of Alexandrov..."

I would argue the Alexandrov work is now already perfectly referenced in the first paragraph and Cai et al.'s strategy of inserting a hormone receptor domain into PQQ-GOx is sufficiently novel and distinct to not constitute the same approach. Perhaps the authors could instead briefly mention (again) the protein engineering work / building blocks as this is IMO an important innovative aspect.

We highly appreciate that the Reviewer acknowledge our approach is distinct and innovative.

(2) page 4 and several places and figures later mention "the apo form of GDH-5E+" but I couldn't find any explanation what is missing between apo and holo enzyme. I am pretty sure, it is the PQQ co-factor we are talking about here but it would be good to spell it out at the first mentioning.

We appreciate the Reviewer pointing out where the manuscript could be more precise. We've clarified in the manuscript that apo-GDH refers to GDH containing no PQQ cofactor.

(3) Legends for Fig 3 and 4 could use a little bit more detail in order to be self-standing. In Fig 3G, the glucose concentration axis running parallel to analyte concentration is probably the final concentration due to repeated addition but that's not entirely clear from "Summary and statistical analysis".

We appreciate the suggestion from the Reviewer. Following their advice, we have provided additional details in the legends for Figures 3 and 4.

Fig. 4C shows glucose crossed out in one section of the graph the meaning of which is not explained (without any glucose addition? Then "4-HT only" or "4HT w/o Glc" would, in my view, be clearer).

We agree with the Reviewer that the crossed-out glucose section is confusing. The graph has been re-labeled accordingly. Thank you for bringing this to our attention.